# Engineered disorder in CO$_2$ photocatalysis

Zhao Li[1,2,3,10], Chengliang Mao[2,10], Qijun Pei[4], Paul N. Duchesne[5], Teng He[4], Meikun Xia[2], Jintao Wang[4], Lu Wang[6], Rui Song[1,2,3], Feysal M. Ali[2], Débora Motta Meira[7,8], Qingjie Ge[4], Kulbir Kaur Ghuman[9]✉, Le He[1,3], Xiaohong Zhang[1,3]✉ & Geoffrey A. Ozin[2]✉

Light harvesting, separation of charge carriers, and surface reactions are three fundamental steps that are essential for an efficient photocatalyst. Here we show that these steps in the TiO$_2$ can be boosted simultaneously by disorder engineering. A solid-state reduction reaction between sodium and TiO$_2$ forms a core-shell c-TiO$_2$@a-TiO$_{2-x}$(OH)$_y$ heterostructure, comprised of HO-Ti-[O]-Ti surface frustrated Lewis pairs (SFLPs) embedded in an amorphous shell surrounding a crystalline core, which enables a new genre of chemical reactivity. Specifically, these SFLPs heterolytically dissociate dihydrogen at room temperature to form charge-balancing protonated hydroxyl groups and hydrides at unsaturated titanium surface sites, which display high reactivity towards CO$_2$ reduction. This crystalline-amorphous heterostructure also boosts light absorption, charge carrier separation and transfer to SFLPs, while prolonged carrier lifetimes and photothermal heat generation further enhance reactivity. The collective results of this study motivate a general approach for catalytically generating sustainable chemicals and fuels through engineered disorder in heterogeneous CO$_2$ photocatalysts.

Amorphous solids may behave as metals[1,2], semiconductors[3,4], insulators[5,6], or superconductors[7,8] and can exhibit electrical, optical, thermal, and mechanical properties distinct from those of their crystalline analogues. Disorder, for example, can introduce both deep and shallow defect states and extended tail states into the band structure of solids, leading to profound changes in the optical and electrical properties of solid-state devices, including those of solar cells[9,10]. The broadly tunable band structure of amorphous metal oxides, together with other merits such as transparency and good stability, has even led them to supersede their crystalline counterparts as promising electron injection layer materials for organic light-emitting diodes[11,12]. In the context of electron mobility, extensive disorder can also impede electron and phonon transport, lowering electrical and thermal conductivity relative to the corresponding ordered phase, thereby making them promising materials for use in thermoelectric power generation and solid-state refrigeration devices[13–15]. These examples demonstrate how engineering the physical properties of solids using disorder enables advances in materials science and technology[16,17]. Little is known, however, about the influence of disorder on surface chemistry and catalysis[18,19], which is the focus of the research described herein.

As a key example, disordered (black) titania is renowned for its enhanced photocatalytic properties relative to its crystalline (white) counterpart[20–23]. Despite great efforts to understand the origin of its reactivity (focusing, for example, on its electronic bandgap[24,25], photon absorption[26,27], and charge-carrier separation kinetics[28,29], pivotal aspects of its surface chemistry remain essentially unknown to

[1]Institute of Functional Nano & Soft Materials (FUNSOM), Soochow University, 199 Ren'ai Road, Suzhou 215123 Jiangsu, PR China. [2]Solar Fuels Group, Department of Chemistry, University of Toronto, 80 St. George Street, Toronto, ON M5S 3H6, Canada. [3]Jiangsu Key Laboratory of Advanced Negative Carbon Technologies, Soochow University, Suzhou 215123 Jiangsu, PR China. [4]Dalian Institute of Chemical Physics, Chinese Academy of Sciences, Dalian 116023 Liaoning, PR China. [5]Department of Chemistry, Queen's University, 90 Bader Lane, Kingston, ON K7L 3N6, Canada. [6]The Chinese University of Hong Kong, Shenzhen, 518172 Shenzhen, Guangdong, People's Republic of China. [7]CLS@APS, Advanced Photon Source, Argonne National Laboratory, Lemont, IL 60439, USA. [8]Canadian Light Source Inc., 44 Innovation Boulevard, Saskatoon, SK S7N 2V3, Canada. [9]Institut National de la Recherche Scientifique, Centre Énergie, Matériaux et Télécommunications, 1650 Boul. Lionel Boulet, Varennes, QC J3X 1S2, Canada. [10]These authors contributed equally: Zhao Li, Chengliang Mao. ✉e-mail: kulbir.ghuman@inrs.ca; xiaohong_zhang@suda.edu.cn; g.ozin@utoronto.ca

this day, due largely to the challenges of defining the surface structure and understanding the physicochemical properties of the amorphous state.

In this article, by means of in situ diffuse reflectance infrared Fourier transform spectroscopy (DRIFTS), solid-state magic angle spinning proton nuclear magnetic resonance ($^1$H-MAS-NMR), electron paramagnetic resonance (EPR), and density functional theory (DFT) modeling of the amorphous state, we present a molecular-level description of surface frustrated Lewis pairs (SFLPs), active for the photocatalytic reduction of $CO_2$ to CO, in a titania heterostructure material comprised of a crystalline core surrounded by an amorphous shell denoted c-$TiO_2$@a-$TiO_{2-x}(OH)_y$.

At the crux of this new kind of SFLP[30], embedded in a disordered a-$TiO_{2-x}(OH)_y$ surface, is the quasi-dative bonding that exists between the Lewis base surface hydroxyl group (OH) and the Lewis acid mixed valence Ti(III,IV) induced by the presence of an oxygen vacancy ([O]) denoted HO-Ti(III)-[O]-Ti(IV), (Fig. 1). Such SFLPs engender high reactivity towards heterolytic $H_2$ dissociation to produce a protonated/hydridic SFLP represented as HOH-Ti-[O]-Ti-H. The capture of $CO_2$, via interaction of the electrophilic carbon of $CO_2$ with the hydride and nucleophilic oxygen of the hydroxide of the SFLP HOH-Ti-[O]-Ti-H site, creates a formic acid (HCOOH) intermediate, which fragments to CO and $H_2O$.

The SFLPs, confined to the surface of the amorphous shell covering the c-$TiO_2$@a-$TiO_{2-x}(OH)_y$ heterostructure, serve multiple functions: they improve the harvesting of visible and near-infrared photons, facilitate the generation and separation of photoexcited charge carriers, prolong charge-carrier lifetimes, and promote local heat generation. Collectively, these effects boost the chemical reactivity of the SFLPs, resulting in a record $CO_2$-to-CO conversion rate of 5.3 mmol $g_{cat}^{-1}$ h$^{-1}$ (350 times superior to that of crystalline c-$TiO_2$) and a turnover frequency of 592 h$^{-1}$, under solar light irradiation of 4.0 W cm$^{-2}$. Thus, the results of this study motivate a general approach to sustainable fuels through engineered disorder in metal oxide-based $CO_2$ photocatalysts.

## Results and discussion
### Na/c-$TiO_2$-$H_2O$ synthesis of oxygen-vacancy- and hydroxyl-laden c-$TiO_2$@a-$TiO_{2-x}(OH)_y$

The core@shell c-$TiO_2$@a-$TiO_{2-x}(OH)_y$ was prepared by ball-milling c-$TiO_2$ (crystalline, 2.2 g) with Na/NaCl mixture (8.8 g) at solid state in Ar atmosphere under ambient pressure. A matrix of excess NaCl was added to facilitate the thorough mixing of the two reactants. Through the mechanochemical-driven redox reaction, mixed oxides with low-valent titanium (Ti(III)) atoms and oxygen vacancies ([O]) were obtained. During this process, we managed to confine the mechanochemistry to a nanometer-thick c-$TiO_2$ surface layer by controlling the ball-milling time.

To understand the charge and stoichiometric balance of the sample preparation, electron paramagnetic resonance (EPR) analysis was employed (Fig. 2a–c). The room-temperature first-derivative spectrum of the reacted mixture showed two major peaks at $g = 2.003$ and $g = 1.977$, which are typical for an unpaired free electron (such as in [O] or Na(0)) and Ti(III), respectively[31–33] (Fig. 2a)). Spectra acquired at 110 K provided better resolution and intensity for these signals, allowing anatase and rutile Ti(III) to be distinguished by $g$ values of (1.990, 1.990, 1.962) and (1.975, 1.975, 1.951), respectively (Fig. 2b). Although the [O] signal in $TiO_2$ typically demonstrates a symmetric Lorentzian line shape, while the Na(0) shows a Dysonian line[34], it is difficult to distinguish the two in the spectra of the reacted mixture due to the complexity of

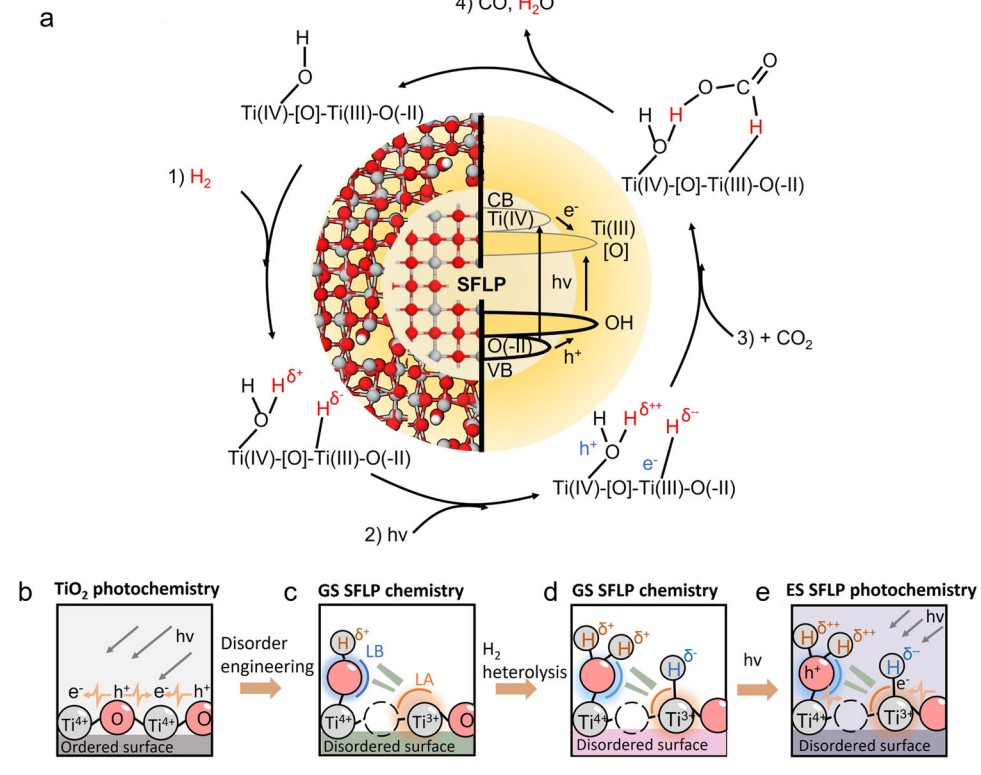

**Fig. 1 | SFLPs, confined to the surface of the amorphous shell of the c-$TiO_2$@-$TiO_{2-x}(OH)_y$ heterostructure, facilitate record photocatalysis $CO_2$ performance. a** Depiction of the central role played by the physicochemical properties in the catalytic hydrogenation of $CO_2$-to-CO by SFLPs. **b–d** Reactive chemistry of an SFLP in c-$TiO_2$@a-$TiO_{2-x}(OH)_y$, highlighting the heterolysis of $H_2$ to form a protonated hydroxyl and a hydridic titanium SFLP. Structures are shown of (**b**) an unmodified crystalline surface, (**c**) formation of an SFLP site, (**d**) a ground-state SFLP site following activation by hydrogen, and (**e**) the enhanced activity excited-state SFLP generated by photoexcitation of the ground-state SFLP.

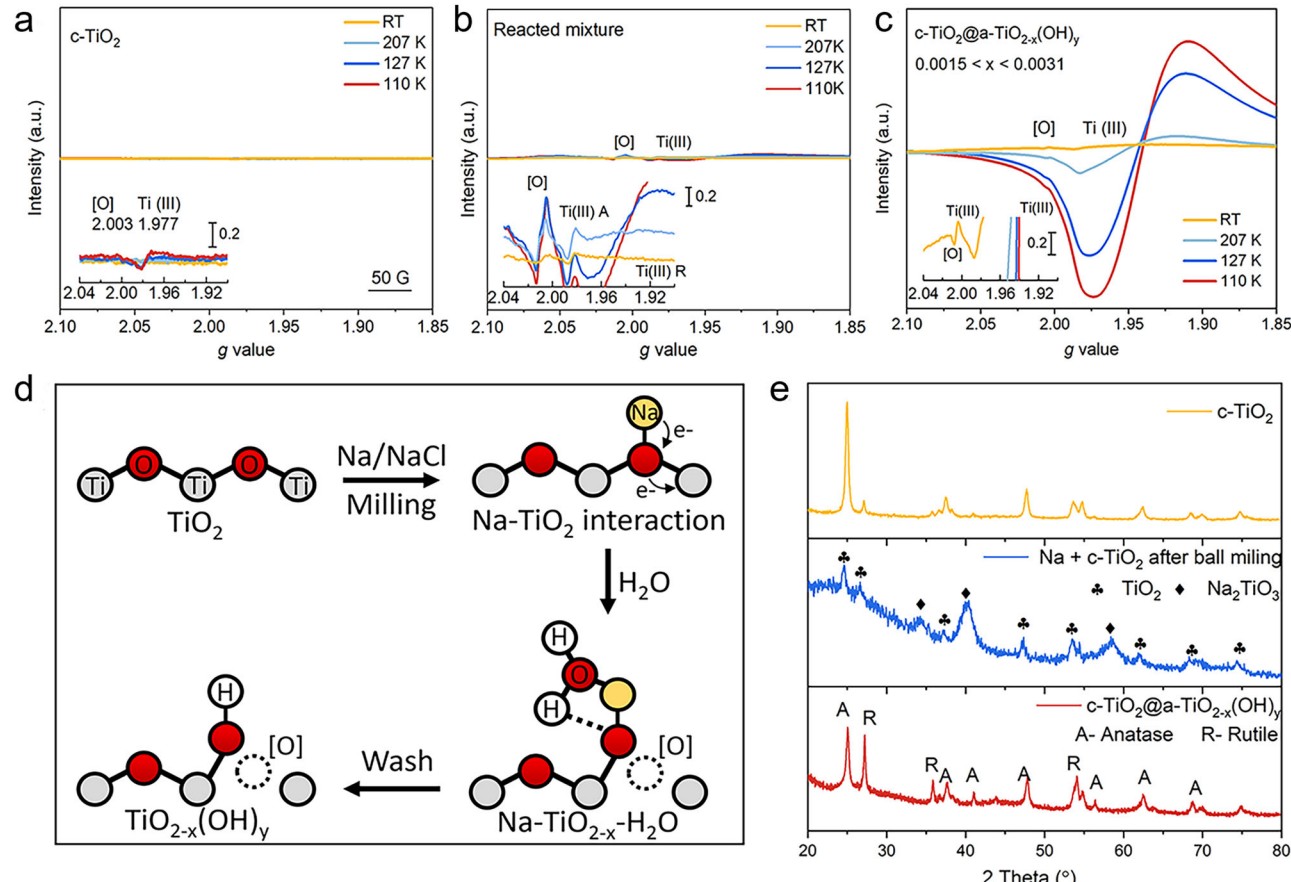

**Fig. 2 | The preparation of c-TiO₂@a-TiO₂₋ₓ(OH)ᵧ from c-TiO₂. a–c** EPR spectra of the pristine c-TiO₂, the mixture of Na + c-TiO₂ + NaCl after ball-milling and the product c-TiO₂@a-TiO₂₋ₓ(OH)ᵧ. **d** Schematic of SFLP formation during c-TiO₂@a-TiO₂₋ₓ(OH)ᵧ synthesis. **e** PXRD patterns of starting c-TiO₂, c-TiO₂@a-TiO₂₋ₓ(OH)ᵧ product, and the intermediate phase.

overlapping peaks (Supplementary Fig. 1). Removing the residuals in the synthesis mixture by thorough washing (by deionized water) helped to resolve this dilemma, resulting in the desired product c-TiO₂@a-TiO₂₋ₓ(OH)ᵧ, the EPR spectra of which showed prominently increased Ti(III) and [O] signals (Fig. 2c). These results evidence that the low-valent Ti(III) and oxygen vacancies were generated mainly during sample washing. Based on this information, the following reaction sequence with balanced equations is proposed to explain the synthesis:

$$TiO_2 + (2x - y)Na \rightarrow Na_{(2x-y)}TiO_2 \qquad (1)$$

$$Na_{(2x-y)}TiO_2/H_2O \rightarrow \{(x - 0.5y)Na_2O + TiO_{2-x+0.5y}\}/H_2O \qquad (2)$$

$$Na_2O + H_2O \rightarrow 2NaOH \qquad (3)$$

$$TiO_{2-x+0.5y} + 0.5yH_2O \rightarrow TiO_{2-x}(OH)_y \qquad (4)$$

Upon the contact of Na and c-TiO₂, Na(0) clusters inserted into the TiO₂ lattice inducing new coordination of Ti(IV)-ONa-Ti(IV). Subsequently, an electron was transferred from Na(0) to Ti(IV) forming a Ti(III) ion, the Ti(III)-ONa-Ti(IV), followed by reacting with water to generate a Ti(III)-[O]-Ti(IV)-OH. The last step is also believed to be the origin of surface disorder in c-TiO₂@a-TiO₂₋ₓ(OH)ᵧ (Fig. 2d). Powder X-ray diffraction (PXRD) confirmed the reaction model, showing evidence of a crystalline intermediate Na₂TiO₃ being formed by ball-milling c-TiO₂ with metallic Na(0), with a decrease in crystallinity for the TiO₂, Fig. 2e. Then, this intermediate was transformed by Na de-intercalation

during the washing process and did not remain in the resulting c-TiO₂@a-TiO₂₋ₓ(OH)ᵧ according to the PXRD and extended X-Ray absorption fine structure (EXAFS) results (Supplementary Fig. 2).

## Ti(III)-[O]-Ti(IV)-OH-related atomic/electronic structure

The thickness of the surface a-TiO₂₋ₓ(OH)ᵧ in the c-TiO₂@a-TiO₂₋ₓ(OH)ᵧ was approximated to be 2 to 6 nm, as indicated by the formation of an amorphous shell/crystalline core heterostructure in the high-resolution transmission electron microscopy (HRTEM) image (Fig. 3a and Supplementary Figs. 3 and 4). HRTEM video and Raman analysis proved that both the c-TiO₂ and the c-TiO₂@a-TiO₂₋ₓ(OH)ᵧ surfaces are stable under measurement conditions, and thus excluded the possibility of the electron beam induced surface amorphization for the c-TiO₂@a-TiO₂₋ₓ(OH)ᵧ (Supplementary Movie 1 and Supplementary Fig. 5). Ti L-edge electron energy loss spectroscopy (EELS) scans taken at several points on a c-TiO₂@a-TiO₂₋ₓ(OH)ᵧ nanoparticle supported this model. Conventional Ti(IV) L₂,L₃-edge features of t₂g-e₂g splitting emerged in the core region, which were similar to those of c-TiO₂, but no such splitting features were observed on the nanoparticle surface, and the L-edge energies shifted to low values by ~2.2 eV (Fig. 3b, c and Supplementary Fig. 6). Given the fact that the t₂g-e₂g splitting stems from the energy difference between Ti 3d-orbitals with different geometric orientations, it is sensitive to the valence of Ti and the presence of point defects in titania[35]. Therefore, the above result suggests a low-valent Ti(III)- and [O]-rich amorphous a-TiO₂₋ₓ(OH)ᵧ shell with Ti-O₆ octahedra distorted and reconstructed to lower symmetry (e.g., distorted Ti-O₅ pyramids).

Furthermore, the X-ray photoelectron spectroscopy (XPS) spectra of c-TiO₂@a-TiO₂₋ₓ(OH)ᵧ showed consistent results. Compared with

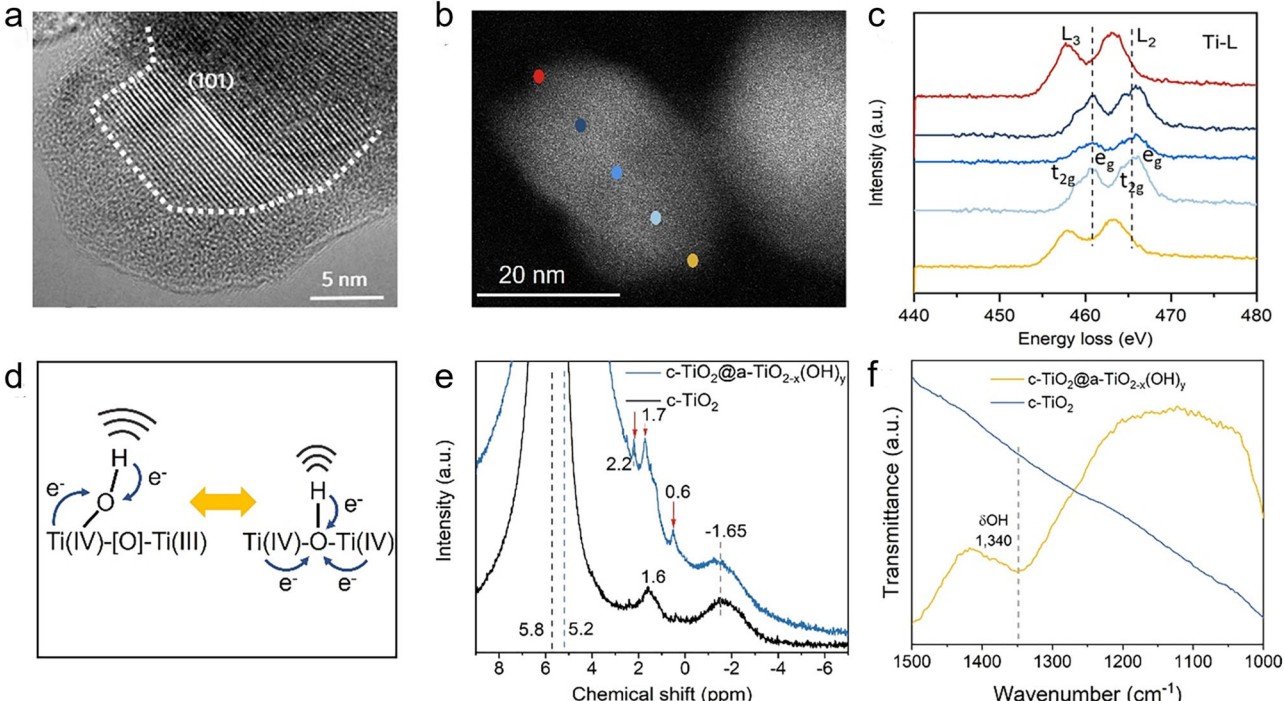

**Fig. 3 | Characterization of crystalline-amorphous heterostructured c-TiO₂@a-TiO₂₋ₓ(OH)ᵧ. a** HR-TEM micrograph of c-TiO₂@a-TiO₂₋ₓ(OH)ᵧ (anatase phase). The amorphous/crystalline interfaces are marked with dotted lines. **b, c** High-angle annular dark-field scanning transmission electron microscopy (HAADF-STEM) image of c-TiO₂@a-TiO₂₋ₓ(OH)ᵧ (**b**) and corresponding Ti-L edge EELS spectra at different positions (**c**). **d** Comparison of geometrical and electronic structures between bridging and terminal hydroxyl. **e, f** ¹H MAS-NMR (**e**) and ATR-FTIR (**f**) spectra of c-TiO₂@a-TiO₂₋ₓ(OH)ᵧ and c-TiO₂.

the c-TiO₂, the Ti 2p spectra of c-TiO₂@a-TiO₂₋ₓ(OH)ᵧ exhibited a slight peak shift from 459.5 eV to 459.3 eV with an emerging shoulder peak at 456.9 eV, signifying the presence of Ti(III) (Supplementary Fig. 7a). The temperature-dependent EPR spectra of c-TiO₂@a-TiO₂₋ₓ(OH)ᵧ showed that the intensity ratio of Ti(III) to [O] decreased from 15.28 at 207 K to 1.716 at room temperature (Fig. 2c), indicating their vicinity enabled an electron shuttle. These results supported our model of mixed-valence [O]-laden Ti(III)-[O]-Ti(IV)-OH with the presence of surface hydroxyl groups, confirmed by a peak at 533.4 eV in the O1s XPS spectra of c-TiO₂@a-TiO₂₋ₓ(OH)ᵧ (Supplementary Fig. 7b).

Theoretically, to obtain the above Ti(III)-[O]-Ti(IV)-OH via the TiO₂₋ₓ₊₀.₅ᵧ + 0.5yH₂O → TiO₂₋ₓ(OH)ᵧ reaction, the H₂O should dissociate on the surface O(-II) and Ti(IV) sites to make charge-balancing Lewis base hydroxide OH(-I) on the Ti(III,IV) sites and protons on the O(-II) as Brønsted acid OH, namely the SFLP. Alternatively, the H₂O can bind at the oxygen vacancy through its oxygen and dissociate to hydroxide and a proton where the latter binds to a lattice oxygen—this reaction pathway generates only Brønsted acid sites on the oxide lattice contrasting with the SFLP model.

To confirm the function of the c-TiO₂@a-TiO₂₋ₓ(OH)ᵧ surface as a SFLP [Ti(III)-[O]-Ti(IV)-OH] instead of a conventional Brønsted acid [Ti(IV)-OH-Ti(IV)], we now focus on distinguishing the two models. Theoretically, the terminal hydroxyl Ti(IV)-OH in the SFLP possesses a higher degree of freedom compared to that of the bridging hydroxyl (Fig. 3d). Thus, their charge density and molecular vibrations should be distinct, which can be monitored through solid-state ¹H magic-angle spinning nuclear magnetic resonance (¹H MAS NMR) and attenuated total reflection Fourier transform infrared (ATR-FTIR) spectroscopy, respectively. The ¹H MAS NMR spectrum of c-TiO₂ showed peaks at 5.8, 1.6, and −1.65 ppm. The peak around 5.8 ppm is typically ascribed to absorbed water, and the peak at 2.2 ppm on neat titanium dioxide is assigned to hydroxyl group[36]. Both water and hydroxyl group are naturally existing species when a metal oxide sample is exposed to air.

The peak at −1.65 ppm appeared to be a background signal as similar signal emerged on the ¹H MAS NMR spectra of c-TiO₂@a-TiO₂₋ₓ(OH)ᵧ as well. Interestingly, the water and hydroxyl peaks of c-TiO₂@a-TiO₂₋ₓ(OH)ᵧ shifted to 5.2 ppm and 1.7 ppm, respectively, and two additional peaks (2.2 and 0.6 ppm) emerged around the hydroxyl peak. Since the chemical shift of ¹H-NMR reflects the shielding effect, compared to c-TiO₂, the lower chemical shift of water on c-TiO₂@a-TiO₂₋ₓ(OH)ᵧ indicated the possible electron donation from oxygen vacancies and/or Ti(III) in a-TiO₂₋ₓ(OH)ᵧ, while the higher chemical shift of hydroxyl (1.7 and 2.2 ppm)[37] suggested the proton of less electron density, which agreed well the Lewis base -OH in the SFLP model. The peak at 0.6 ppm can be assigned to several species[38], including terminal hydroxyl group and proton bonded to oxygen vacancy, which are closely related to the amorphous a-TiO₂₋ₓ(OH)ᵧ shell (Fig. 3e). Furthermore, ATR-FTIR spectra of c-TiO₂@a-TiO₂₋ₓ(OH)ᵧ demonstrated a 1340 cm⁻¹ peak, which is likely the bending mode of terminal OH[39] induced by the symmetry-breaking of a bridging OH (Fig. 2d and Supplementary Fig. 8). Additionally, new peaks in the 3610-3680 cm⁻¹ region, belonging to the stretching mode of terminal OH[39,40], appeared on the ATR-FTIR spectra of c-TiO₂@a-TiO₂₋ₓ(OH)ᵧ, while that of c-TiO₂ only showed the bridging OH feature located in the 3690-3750 cm⁻¹ region (Supplementary Fig. 8). Despite the OH stretching peaks being low in intensity compared to the OH bending peaks, it was quite clear that the terminal OH species increased with the introduction of disorder TiO₂₋ₓ(OH)ᵧ on the c-TiO₂ surface. Both theoretic and experimental results confirm the SFLP feature of c-TiO₂@a-TiO₂₋ₓ(OH)ᵧ, where mixed-valence Ti(III, IV), [O], and terminal OH are in close proximity, forming the Ti(III)-[O]-Ti(IV)-OH.

## SFLPs-related electronic properties, light absorption, and charge carrier separation

The reduction extent of titania during sample preparation determined the degree of non-stoichiometry (x) and surface disorder as well as the

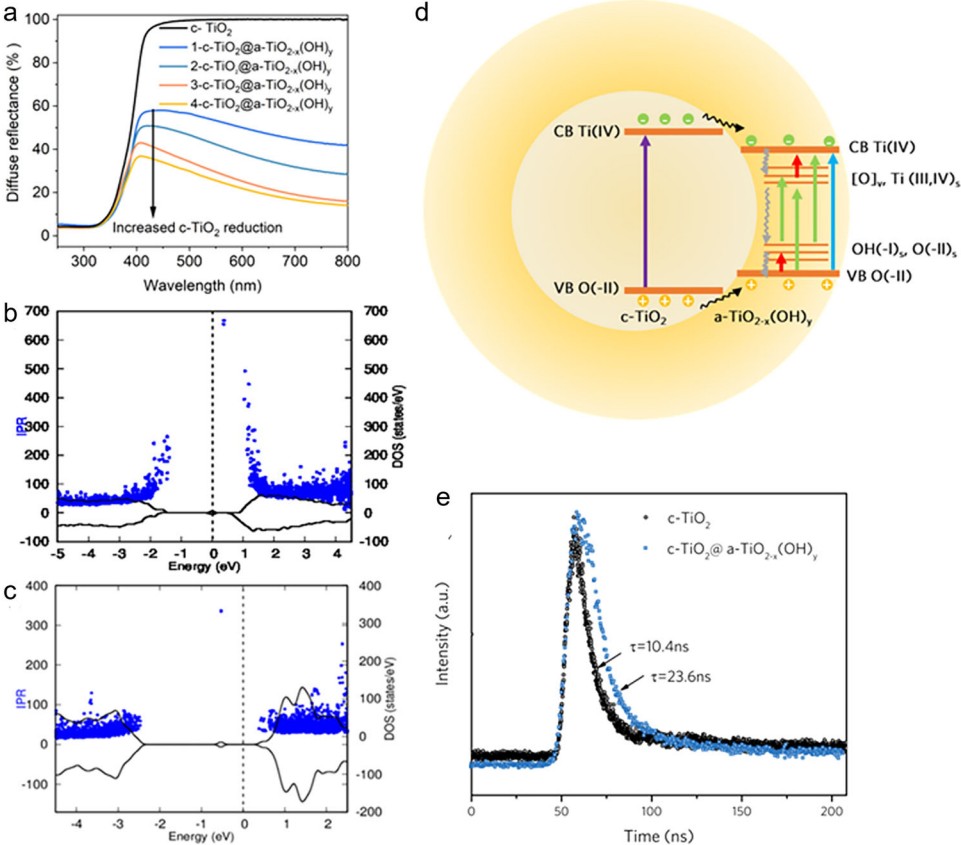

**Fig. 4 | Opto-electronic properties of c-TiO$_2$@a-TiO$_{2-x}$(OH)$_y$. a** UV–Vis-NIR diffuse reflectance spectra (DRS) of c-TiO$_2$@a-TiO$_{2-x}$(OH)$_y$ as a function of reduction degree. The x was quantified in each sample. **b, c** The total density of states (black line, right axis) and the corresponding values of the inverse participation ratio (IPR) (blue dots, left axis) for a-TiO$_{2-x}$(OH)$_y$ (**b**) and crystalline TiO$_{2-x}$ (**c**) surfaces.

**d** Schematic of charge carrier separation and transfer pathways in c-TiO$_2$@a-TiO$_{2-x}$(OH)$_y$. The colors of arrows indicate the wavelength of incident light. **e** Time-resolved photoluminescence spectroscopy decay curves of c-TiO$_2$ and c-TiO$_2$@a-TiO$_{2-x}$(OH)$_y$.

population of Ti(III), oxygen vacancies, and OH sites, which further corresponds to their optical spectra and band gaps[41]. The optical reflectance spectrum (UV-Vis-NIR) of c-TiO$_2$ demonstrated UV light absorption with a band edge at ~380 nm, while that for c-TiO$_2$@a-TiO$_{2-x}$(OH)$_y$ (0.0015 < x < 0.0031) demonstrated strong broadband light absorption (Fig. 4a). Increasing the x value from 0.0002 < x < 0.0004 to 0.0015 < x < 0.0031 in c-TiO$_2$@a-TiO$_{2-x}$(OH)$_y$ samples (1→4) showed the evolution of Vis-NIR absorption, and red shifting of the absorption edges (Fig. 4a), corresponding to bandgaps of 3.49, 3.46, 3.43 and 3.38 eV (Supplementary Fig. 9), respectively; all of these bandgap energies were smaller than the value of 3.57 eV for pristine c-TiO$_2$, indicating the increased sunlight harvest of c-TiO$_2$@a-TiO$_{2-x}$(OH)$_y$ compared with c-TiO$_2$. The detailed band-edge positions of c-TiO$_2$@a-TiO$_{2-x}$(OH)$_y$ was provided by ultraviolet photoelectron spectroscopy (UPS). Valence band ($E_{VB}$) positions of c-TiO$_2$ and a-TiO$_{2-x}$(OH)$_y$ (0.0015 < x < 0.0031) were found at −7.7 and −7.59 eV (vs. vacuum), and the corresponding conduction band ($E_{CB}$) positions were found at −4.13 and −4.21 eV (vs. vacuum), respectively (Supplementary Fig. 10).

Density of states (DOS) and the delocalization degree of electrons were then simulated computationally in both the amorphous and crystalline components to provide mechanistic insights. Specifically, inverse participation ratio (IPR) analysis was conducted to investigate charge mobility and the localization of energy states on amorphous TiO$_{2-x}$(OH)$_y$ and crystalline TiO$_{2-x}$ surfaces. In Fig. 4b, c, the IPR and DOS of amorphous TiO$_{2-x}$(OH)$_y$ and crystalline TiO$_{2-x}$ are depicted, where a large IPR value represents highly localized states and a small IPR value represents delocalized states. The IPR results showed that

the Ti 3d in the bottom of the CB (conduction band) and O 2p states at the top of the VB (valence band) for amorphous TiO$_{2-x}$(OH)$_y$ are strongly localized as compared to crystalline TiO$_{2-x}$. Specifically, the OH and [O] vacancy resulted in shallow band states near the VB edge and the [O] vacancy resulted in shallow band states near the CB edge, respectively (Supplementary Fig. 11). These results indicate the high-concentration self-trapped polarons and excitons of a-TiO$_{2-x}$(OH)$_y$ compared to crystalline TiO$_{2-x}$, which facilitate charge carrier separation by trapping photo-generated holes and electrons by SLFPs OH and Ti(III), respectively.

Based on CB and VB positions obtained from the UPS experiment, DOS and charge localization analyses via DFT, a schematic illustration for electron-hole transfer, separation, and recombination, is proposed in Fig. 4d. The UV light can excite both the core c-TiO$_2$ and the shell a-TiO$_{2-x}$(OH)$_y$ because the photon energy of the UV light (λ < 347 nm or E > 3.57 eV) is larger than the bandgap energy of c-TiO$_2$ ($E_g$ = 3.57 eV) or a-TiO$_{2-x}$(OH)$_y$ ($E_g$ = 3.38 eV). The calculated penetration depth ($D_p$) of UV light for TiO$_2$ further supported the above results (Supplementary Figs. 12 and 13). c-TiO$_2$ and a-TiO$_{2-x}$(OH)$_y$ demonstrate similar UV-absorbing capacities, thus showing a similar penetration depth in UV region. Given the $D_p$ is larger than the 3–6 nm thickness of the shell, the UV light will be partially absorbed by surface a-TiO$_{2-x}$(OH)$_y$ and then reach the core c-TiO$_2$ in the c-TiO$_2$@a-TiO$_{2-x}$(OH)$_y$. Therefore, both the core c-TiO$_2$ and shell a-TiO$_{2-x}$(OH)$_y$ will be excited by the UV light to yield photo electrons and holes. Simultaneously, the visible light penetration depth of black titania is within several micrometers, so the visible light cannot be fully absorbed by a 3–6 nm surface a-TiO$_{2-x}$(OH)$_y$ layer but will be absorbed by the sub-surface catalyst

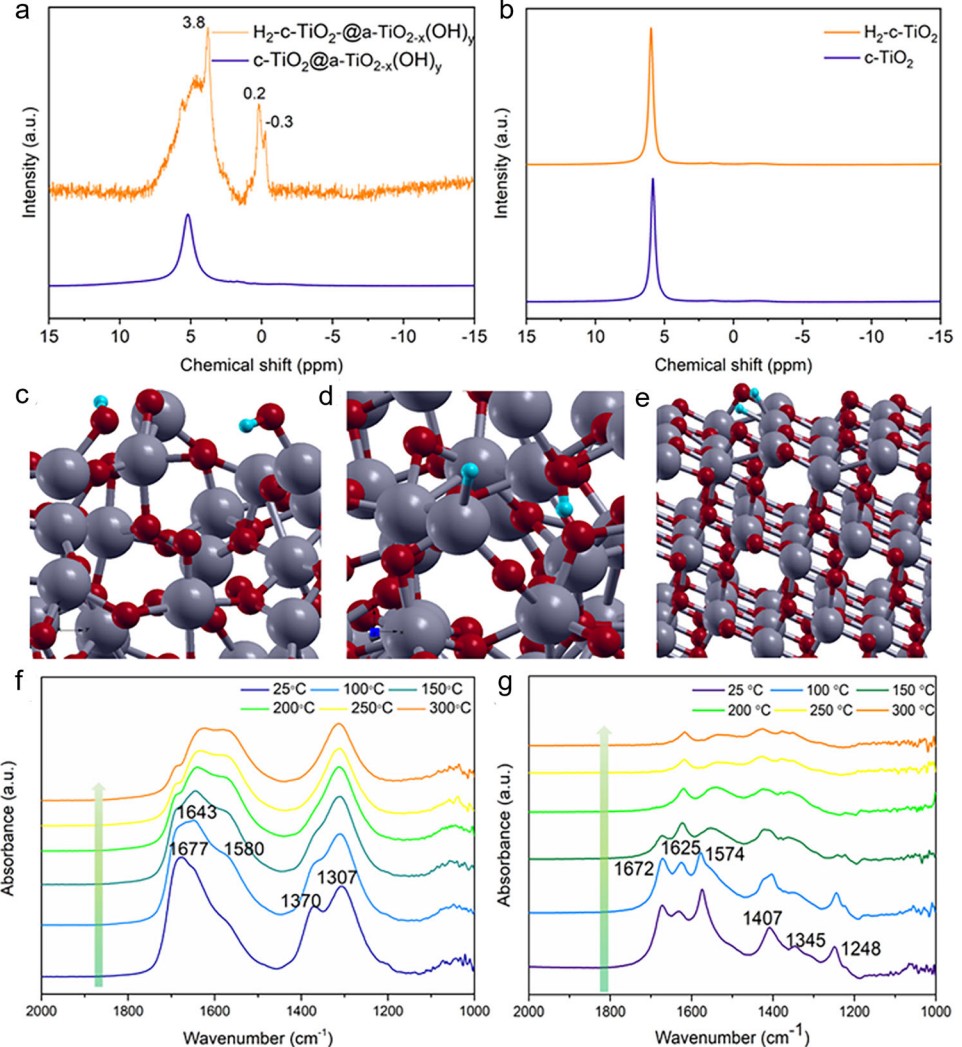

**Fig. 5 | Activation of CO₂ by SFLPs on c-TiO₂@a-TiO₂₋ₓ(OH)ᵧ. a, b** Ex situ solid-state ¹H MAS-NMR spectra of c-TiO₂@a-TiO₂₋ₓ(OH)ᵧ (a) and c-TiO₂ (b) before (purple) and after (orange) exposure to H₂ at atmospheric pressure and room temperature. **c–e** H₂ adsorption resulting in the formation of two surface OH (c) or a Ti·H and an OH on amorphous titania surfaces (d), or absorbed H atoms on crystalline TiO₂₋ₓ (e). The Ti, O and H atoms are represented in gray, red, and blue, respectively. **f, g** In situ DRIFTS spectra of CO₂ reduction over c-TiO₂@a-TiO₂₋ₓ(OH)ᵧ (f) and c-TiO₂ (g) at various temperatures while flowing 2 sccm H₂, 2 sccm CO₂, and 16 sccm He.

particles with a stacking thickness of several micrometers. Accordingly, the visible and IR light can excite electrons on the defect- and OH-related energy levels of the a-TiO₂₋ₓ(OH)ᵧ. Because the CB of a-TiO₂₋ₓ(OH)ᵧ is lower than that of c-TiO₂, the photogenerated electrons on the c-TiO₂ preferentially transfer across the crystalline-amorphous interface to reach the a-TiO₂₋ₓ(OH)ᵧ and get trapped by an unsaturated Ti(III) and/or oxygen-vacancy sites. Therefore, the crystalline core-amorphous shell structure facilitates the charge carrier spatial separation, which enhances the excited-state Lewis basicity and acidity of the Ti(III)-[O]-Ti(IV)-OH, namely the ES SFLP, by the trapped electrons and holes. This ES SFLP not only enhances chemical reactivity compared to its GS counterpart but also possibly prolongs the lifetime of the photogenerated electrons and holes during the H₂ and CO₂ reaction.

Inspired by above calculation, time-resolved photoluminescence (PL) spectroscopy was performed at room temperature to testify the diffusion and recombination processes of photogenerated electron-hole pairs in c-TiO₂@a-TiO₂₋ₓ(OH)ᵧ. All PL decay curves exhibited extremely fast relaxation (Fig. 4e) and were fitted by a bi-exponential model. The pristine TiO₂ gave an average exciton-decay time of 10.4 ns, while a longer exciton-decay time of 23.6 ns was observed for the c-TiO₂@a-TiO₂₋ₓ(OH)ᵧ (Supplementary Table 1). This supports the proposed model (depicted in Fig. 4d) of charge transfer between the c-TiO₂ core and the a-TiO₂₋ₓ(OH)ᵧ shell, prolonging the lifetime of charge carriers for subsequent photoreaction.

## Activation of CO₂ and H₂ by SFLPs on c-TiO₂@a-TiO₂₋ₓ(OH)ᵧ

Previous studies evidenced a key feature of the frustrated Lewis pair as triggering H₂ activation[42] under ambient temperatures. The obtained products are a proton and a hydride, of which the latter shows high chemical reactivity for CO₂ reduction. Therefore, to check the chemical function of the SFLP in the c-TiO₂@a-TiO₂₋ₓ(OH)ᵧ, its reaction with H₂ was monitored through the NMR. Solid state ¹H MAS NMR of c-TiO₂@a-TiO₂₋ₓ(OH) showed a major peak at *ca.* 5.2 ppm which was typically ascribed to Ti-OH₂ species[43], and peaks between −1.6—2.2 ppm with negligible intensities compared to that at 5.2 ppm. After exposure to H₂ at room temperature, a protonated Ti-OH₂ peak at 3.8 ppm, and two intense hydridic Ti·H peaks[44,45] at 0.14 ppm & −0.2 ppm emerged, signifying the heterolytic dissociation of H₂ on SFLPs (Fig. 5a). In contrast, there was no significant change in the NMR spectra of c-TiO₂ before and after introducing H₂ (Fig. 5b).

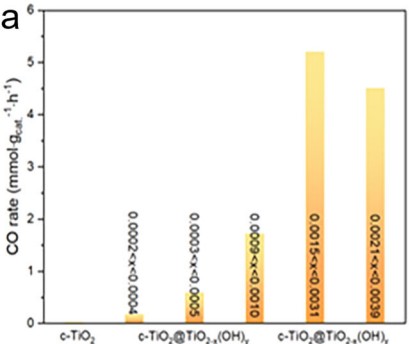
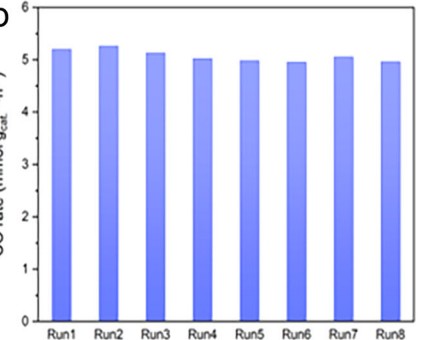
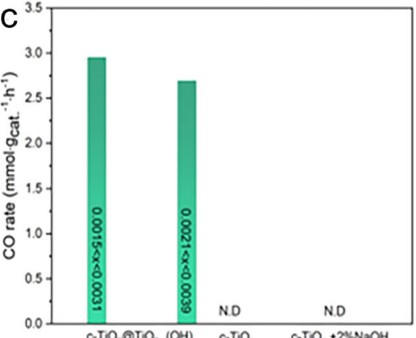

**Fig. 6 | Photocatalytic performance of oxides in the batch reactor without external heating. Initial reactants were 15 psi of $CO_2$ and 15 psi of $H_2$ (total pressure of *ca.* 2 bar). a** Non-stoichiometry dependent CO production rates of the c-$TiO_2$@a-$TiO_{2-x}(OH)_y$ under full-spectrum Xe light, 4.0 W cm$^{-2}$. **b** Stability of c-$TiO_2$@a-$TiO_{2-x}(OH)_y$ under full-spectrum Xe light (4.0 W cm$^{-2}$) in 8 h. **c** CO production rates of c-$TiO_2$@a-$TiO_{2-x}(OH)_y$, pristine c-$TiO_2$ and 2% Na/c-$TiO_2$ under visible light (2.8 W cm$^{-2}$).

Theoretical calculations on the amorphous structure were then conducted to provide mechanistic understanding of the origin of the facile $H_2$ heterolysis over the SFLP site of c-$TiO_2$@a-$TiO_{2-x}(OH)_y$. The interaction of $H_2$ with amorphous $TiO_2$ (a-$TiO_2$) was screened on three Ti sites (Ti-4c, Ti-5c, and Ti-6c) and ten O sites (a to i) via geometry optimization (Supplementary Fig. 14). In all cases (except sites 'i' and 'Ti-4c'), $H_2$ adsorption on surface was exothermic but only resulted in the formation of surface OH, as shown in Fig. 5c. Surprisingly, the introduction of O vacancies on amorphous surfaces resulted in facile exothermic $H_2$ heterolysis, with $\delta E_{ads} = -0.06$ Ry and the generation of a surface OH and a Ti-H, Fig. 5d. Bader charge analysis showed that the hydrogen in the OH bore a charge of +0.99e while −0.6e in the hydride (Supplementary Table 2), bonding to weakly non-coordinative Lewis basic O (−1.7e) and Lewis acidic Ti (+2.1e), respectively. In contrast, the $H_2$ could not dissociate to form OH when it interacted with the crystalline c-$TiO_{2-x}$ surface having vacancies at O-2c (Fig. 5e) and O-3c sites (Supplementary Fig. 15).

Subsequently, in situ DRIFTS was applied to monitor the $CO_2$ reduction over the SFLP-laden c-$TiO_2$@a-$TiO_{2-x}(OH)_y$ at molecular level. In the presence of both $H_2$ and $CO_2$, three major C-related peaks emerged at 1307 cm$^{-1}$, 1370 cm$^{-1}$, and 1677 cm$^{-1}$ (Fig. 5f), which could be assigned as carboxylate species (1307 cm$^{-1}$)[46,47] and formate (1370 cm$^{-1}$ and 1677 cm$^{-1}$)[48,49]. Such a formate species agreed well with the reaction model of $CO_2$ addition to the protonated and hydridic SFLPs by its nucleophilic O(-II) and electrophilic C(IV), respectively (Fig. 1). Consistently, the peak intensities of formate species decreased at high temperature (200 → 300 °C) as they were converted to CO, with the increased intensity at 1643 cm$^{-1}$ corresponding to the $H_2O$ product induced scissor modes. By contrast, carbonates (1345 cm$^{-1}$, 1625 cm$^{-1}$, and 1574 cm$^{-1}$) and bicarbonates (1248 cm$^{-1}$, 1407 cm$^{-1}$, and 1672 cm$^{-1}$)[48,50,51] were generated on the surface of c-$TiO_2$ (Fig. 5g). The distinct surface chemistry between c-$TiO_2$ and amorphous $TiO_{2-x}(OH)_y$ in RWGS reaction was also confirmed by the apparent activation energy ($E_a$). Arrhenius plots of temperature-dependent CO rates suggest an $E_a$ of 75.40 kJ mol$^{-1}$ for c-$TiO_2$@a-$TiO_{2-x}(OH)_y$, which decreased by ~17% compared with 90.27 kJ mol$^{-1}$ of c-$TiO_2$ (Supplementary Fig. 16).

### Photocatalytic performance

With the favorable band structure and surface reactivity, the SFLPs-laden c-$TiO_2$@a-$TiO_{2-x}(OH)_y$ (0.0015 < x < 0.0031) demonstrated $CO_2$ photocatalytic activity of 5.3 mmol g$_{cat}$$^{-1}$ h$^{-1}$ under solar light irradiation (4.0 W cm$^{-2}$), which was 350 times that of the pristine c-$TiO_2$ (Fig. 6a). The turnover frequency (TOF) of c-$TiO_2$@a-$TiO_{2-x}(OH)_y$ was as high as 592 h$^{-1}$, superb among known catalysts tested under comparable photocatalytic conditions[52] (Supplementary Table 3). The

apparent quantum yield (AQY) of the full Xe lamp spectrum can achieve 0.09% (Supplementary Note). This is likely a lower limit as neither the photocatalyst nor photoreactor architectures have been engineered and optimized for high photon capture efficiency, the key to high energy efficiency of the integrated system.

Notably, the c-$TiO_2$@a-$TiO_{2-x}(OH)_y$ (0.0015 < x < 0.0031) demonstrated no apparent activity decrease in eight consecutive runs in the batch reactor (Fig. 6b) or a 48-h on-stream reaction under 300 °C in a flow reactor (Supplementary Fig. 17). The structural stability of spent c-$TiO_2$@a-$TiO_{2-x}(OH)_y$ was supported by PXRD and XPS, which showed no significant changes compared to that before reaction (Supplementary Figs. 18 and 19).

Furthermore, using an isotopically labeled $^{13}CO_2$-$H_2$ feedstock gave $^{13}CO$ as the only product for c-$TiO_2$@a-$TiO_{2-x}(OH)_y$, confirming the carbon source from the $CO_2$ feedstock (Supplementary Fig. 20).

ICP-OES analysis of c-$TiO_2$@a-$TiO_{2-x}(OH)_y$ suggested a 2 wt% residual Na, while our control sample prepared by loading 2 wt% Na (NaOH) on c-$TiO_2$ demonstrated 0.19 mmol g$_{cat}$$^{-1}$ h$^{-1}$ activity, a negligible increase compared with the pristine sample, excluding the possibility that Na-dominated activity enhancement of the c-$TiO_2$@a-$TiO_{2-x}(OH)_y$ sample (Supplementary Fig. 21). These results support the SFLP-laden c-$TiO_2$@a-$TiO_{2-x}(OH)_y$ (0.0015 < x < 0.0031) as a promising $CO_2$ photocatalyst.

Interestingly, the aforementioned synergy effect between the c-$TiO_2$ core and the a-$TiO_{2-x}(OH)_y$ shell, as shown in Fig. 4d, could also be reflected in the activity by varying the non-stoichiometry of the c-$TiO_2$@a-$TiO_{2-x}(OH)_y$ catalyst. To amplify, the CO production rate of c-$TiO_2$@a-$TiO_{2-x}(OH)_y$ samples first increased from 0.17 mmol g$_{cat}$$^{-1}$ h$^{-1}$ (0.0002 < x < 0.0004) to 5.3 mmol g$_{cat}$$^{-1}$ h$^{-1}$ (0.0015 < x < 0.0031), then decreased to 4.7 mmol g$_{cat}$$^{-1}$ h$^{-1}$ by further increasing the non-stoichiometry to 0.0021 < x < 0.0039, demonstrating a volcano activity-non-stoichiometry relationship (Fig. 6a, Supplementary Fig. 22). HRTEM images evidenced the optimal sample (0.0015 < x < 0.0031) with an amorphous shell of 3–6 nm in thickness, while the samples of 0.0002 < x < 0.0004 and 0.0021 < x < 0.0039 showed negligible and ~10 nm shell thickness, respectively (Supplementary Fig. 23). Given the efficiency of a photocatalytic reaction is determined by the efficiencies of light absorption, charge separation and finally the surface reaction, above results suggested the synergy effect resulting from a tradeoff between 1) the high charge-carrier separation efficiency and high photo-available surface area related to the core (c-$TiO_2$) and 2) the high surface reactivity induced by the SFLP shell (a-$TiO_{2-x}(OH)_y$) (Supplementary Fig. 13).

In addition, wavelength-dependent activity tests further shed light on the contribution of the core-shell synergy effect in the c-$TiO_2$@a-$TiO_{2-x}(OH)_y$ (0.0015 < x < 0.0031). After deactivating the core c-$TiO_2$ by

filtering the UV part ($\lambda < 420$ nm) of the Xe lamp, the c-$TiO_2$@a-$TiO_{2-x}(OH)_y$ sample exhibited visible light-driven activity of 3.0 mmol CO $g_{cat}^{-1}$ $h^{-1}$ (Fig. 6c). By comparing the UV-corresponded activity [(5.3–3.0) mmol $g_{cat}^{-1}$ $h^{-1}$] with that of the pure c-$TiO_2$ (0 mmol $g_{cat}^{-1}$ $h^{-1}$), the contribution of the synergy effect is 43% of the full-spectrum activity (2.3 vs. 5.3 mmol $g_{cat}^{-1}$ $h^{-1}$), which is the upper limit. While the lower limit is estimated by comparing the UV activity (2.3 mmol $g_{cat}^{-1}$ $h^{-1}$) with that of the $TiO_2$@a-$TiO_{2-x}(OH)_y$ sample bearing the highest x value ($0.0021 < x < 0.0039$; 1.9 mmol $g_{cat}^{-1}$ $h^{-1}$), which is 8% [(2.3–1.9) vs. 5.3 mmol $g_{cat}^{-1}$ $h^{-1}$]. In the Vis-IR region devoid of the synergy effect, wavelength-dependent CO production rate agreed well with the light-absorbing capability as reflected by DRS spectra (Supplementary Fig. 24). The photothermal contribution to the activity was identified by changing the $CO_2/H_2$ gas ratio in the reactor from 1:1 to 5:1 without varying the light intensity (Supplementary Fig. 25). The CO production rate doubled to 11.2 mmol $g_{cat}^{-1}$ $h^{-1}$ due to the better insulating effect of $CO_2$ relative to $H_2$. ASPEN Plus was also employed to calculate the local temperature of c-$TiO_2$@a-$TiO_{2-x}(OH)_y$ using a $CO_2/H_2$ ratio of 5:1. The predicted temperature reached as high as 200 °C (Supplementary Table 4).

Intentionally engineered SFLPs in the amorphous shell of c-$TiO_2$@a-$TiO_{2-x}(OH)_y$ boost the harvesting potential of solar photons and the effectiveness of the photothermal effect, while the heterostructure benefits the generation, separation, and lifetime of electron-hole pairs on the photo reactivity of HOTi-[O]-Ti SFLPs towards $H_2$ heterolysis and $CO_2$ reduction. The paradigm of integrating disorder engineering with surface frustrated Lewis pairs in a core-shell crystalline-amorphous c-$TiO_2$@a-$TiO_{2-x}(OH)_y$ heterostructure provides a new paradigm for designing and implementing photocatalysts for synthesizing sustainable fuels from carbon dioxide feedstock.

## Methods

### Material synthesis
All chemicals used were of analytical grade and used without any further purification. Commercial P25(c-$TiO_2$) (99%) was purchased from Alfa Aesar.

The Na/NaCl mixture, composed of small Na particles dispersed within NaCl, is expected to be an effective agent for the reduction of metal oxides. The weight ratio between Na and NaCl is about 1:10 and the milling speed is 150–200 rpm. Following a typical experiment for synthesizing c-$TiO_2$@a-$TiO_{2-x}(OH)_y$, P25 was milled with Na/NaCl powder in a weight ratio of 1:4 under argon atmosphere at room temperature using a Retsch PM400 planetary ball mill, at a milling rate of 180 rpm. All above procedures were conducted in a glovebox filled with Ar. The samples (4-c-$TiO_2$@a-$TiO_{2-x}(OH)_y$) were milled for 4 h before being collected and washed several times with deionized water to remove the residual Na/NaCl. Finally, the samples were vacuum-dried at room temperature to isolate the final, dark-blue a-$TiO_{2-x}(OH)_y$@c-$TiO_2$ powder. All characterized and tested c-$TiO_2$@a-$TiO_{2-x}(OH)_y$ samples were washed 3 times with Na content of about 2 wt% Na if not mentioned or emphasized the wash times. Other samples with different oxygen-vacancy concentrations (1- c-$TiO_2$@a-$TiO_{2-x}(OH)_y$, 2- c-$TiO_2$@a-$TiO_{2-x}(OH)_y$ and 3-c-$TiO_2$@a-$TiO_{2-x}(OH)_y$) were prepared by milling P25 with Na/NaCl powder in a weight ratio of 1:1 at a milling rate of 150 rpm for 4 h, milling with Na/NaCl powder in a weight ratio of 1:1 at a milling rate of 180 rpm for 4 h, and milling with Na/NaCl powder in a weight ratio of 1:2 at a milling rate of 180 rpm for 4 h, respectively.

### Materials characterization
EPR measurements were conducted on a Bruker EMXmicro-6/1 EPR spectrometer. The powdered samples were packed into a Wilmad EPR quartz tube and closed with a cap. At least two co-added scans were performed for each sample (receiver gain: 30 dB) using a 1000.0 G sweep width of 60.0 s (or a 1500.0 G sweep width of 90.0 s), and a microwave power of ~5.0 mW (attenuation: 16 dB). The DPPH was used

as a standard to calibrate the central field every time before the sample measurements. Low-temperature experiments were enabled by liquid nitrogen. PXRD was performed on a Bruker D2-Phaser X-ray diffractometer, using Cu Kα radiation at 30 kV. XPS was performed in an ultrahigh vacuum chamber with base pressure of $10^{-9}$ torr. This system used a Thermo Scientific K-Alpha XPS spectrometer with an Al Kα X-ray source (1,486.7 eV) operating at 12 kV and 6 A. The spectra were obtained using an analyzer pass energy of 50 eV with a resolution of 0.1 eV. HR-TEM images were acquired on an aberration-corrected FEI Titan 380–300 microscope operated at a 300 kV accelerating voltage. HR-TEM images were also evaluated by calculating their 2D Fourier transform, yielding information regarding their crystal lattices. ICP-OES was carried out on a Thermo Scientific iCAP 7000 Series ICP Spectrometer. The surface areas of samples were measured on an Autosorb-1 system (Quantachrome, USA) from $N_2$ adsorption isotherms obtained via the BET method. UV-visible diffuse reflectance spectra were obtained for dry-pressed disk samples using a Cary 500 Scan Spectrophotometer (Varian, USA) over a range of 200 to 800 nm. $BaSO_4$ was used as a reflectance standard in the UV-visible diffuse reflectance experiment. The photoluminescence (PL) spectra were measured on an Andor Shamrock SR-750 fluorescence spectrometer with a Xe-lamp as an excitation source (Andor Technology Ltd, Belfast, UK). A CCD detector combined with a monochromator was used for signal collection. The $^1H$ solid state MAS NMR spectra were obtained at a spinning rate of 20 kHz. The NMR Spectra were calibrated to reference adamantane with optimized parameters: pulse width (pwX90) = 3.45 microseconds, fine power (aX90) = 2700, course power (tpwr) = 59, and synthesizer offset (tof) = 1192.9. Number of scans = 64, delay time = 6 s. Samples for $^1H$ solid state MAS NMR were treated by $H_2$ and then transferred to a glovebox with an Ar atmosphere for sample loading. The overall local temperature of the sample in a high-intensity batch reactor ($CO_2/H_2$ ratio = 5:1, 40 suns illumination) was estimated by allowing the system to approach equilibrium and comparing the final composition with ASPEN Plus V9 output, using the Peng-Robinson property package with the Gibbs reactor.

### Measurements of the gas-phase photocatalytic reduction $CO_2$
For gas-phase photocatalytic testing in a batch reactor, the samples were prepared by drop-casting samples, dispersed in DI water, onto binder-free borosilicate glass microfiber filters with an area of 1 $cm^2$ and then drying them under ambient conditions prior to reactor testing. Gas-phase photocatalytic experiments were conducted in a custom-fabricated 11.8 mL stainless steel batch reactor with a fused silica view port sealed with a Viton O-ring. The reactor was evacuated using an Alcatel dry pump before being purged with $H_2$ (99.9995%) at a flow rate of 15 sccm. The pressure inside the reactor was monitored during the reaction using an Omega PX309 pressure transducer. After $H_2$ purging, the reactor was infiltrated with $H_2$ and $CO_2$ gas, to achieve the desired pressure ratio, before being sealed. The reactor was irradiated using a 300 W Xe lamp at 4.0 W $cm^{-2}$ light intensity for a duration of 1 h per run. For the tests using a $CO_2/H_2$ gas ratio of 1:1, the reactor was infiltrated with 15 psi $CO_2$ and 15 psi $H_2$. For the tests using a $CO_2/H_2$ gas ratio of 5:1, $CO_2$ was infiltrated until pressures of 25 psi $CO_2$ and 5 psi $H_2$ were reached. Visible light with an intensity of 2.8 W $cm^{-2}$ was obtained using a 420 nm optical cut-off filter to remove UV light from the Xe source. Product gases were analyzed using flame ionization and thermal conductivity detectors installed in an SRI-8610 gas chromatograph equipped with 30 Mole Sieve 13a and 60 Haysep D column. Isotope product gases were measured using an Agilent 7890 A gas chromatographic mass spectrometer with a 60 m GS-Carbon PLOT column fed to the mass spectrometer. The hydrogenation experiments in a flow system were conducted in a tubular quartz reactor with an inner diameter of 2 mm, in which ~20 mg of catalyst sample was packed and irradiated using an unfiltered 300 W Xe lamp. The diameter of the light spot was ~2 cm, with an area of about 3.14 $cm^2$, which

could fully cover the sample. An OMEGA temperature controller was attached to a heating cartridge inserted into the copper block, along with a thermocouple inserted into the quartz tube (in contact with the catalyst bed) for control of the catalyst temperature.

## In situ DRIFTS measurement

The in situ DRIFTS experiments were performed on a Thermo Scientific Nicolet iS50 FTIR Spectrometer with a mercury cadmium telluride detector cooled with liquid nitrogen. The spectrometer was equipped with a Harrik Praying Mantis diffuse reflection accessory and a Harrick high-temperature reaction chamber. The temperature was controlled with a Harrick ATC-024-3 Temperature Controller. Fresh reduced oxide sample was loaded onto a sampling accessory with a diamond-coated zinc selenide window and then sealed in the reactor. The reactor was heated to 300 °C and kept for 2 h in He (20 sccm) to remove the moisture and surface contaminants on the sample. After cooling the reactor down to 25 °C in He, the IR spectrum was collected to provide the baseline for subsequent differential spectra. The spectra were collected at 25 °C in $H_2$ (5 sccm) and He (15 sccm) within 0 to 20 min. The $H_2$ was then turned off while He (15 sccm) was kept purging at 25 °C for 30 min to remove residual $H_2$ within the reactor. Afterward, the spectra were collected under flowing $H_2$ (2 sccm), $CO_2$ (2 sccm), and He (16 sccm) at various temperatures (25 °C, 100 °C, 150 °C, 200 °C, 250 °C, and 300 °C). Each temperature was held for 20 min.

## Computational details

The amorphous $TiO_2$ (a-$TiO_2$) model was prepared via the melting and quenching method as demonstrated elsewhere[53]. The properties of the prepared a-$TiO_2$ model were in agreement with experimental and theoretical data available. The structural analysis of these models suggests that local structural features of bulk crystalline $TiO_2$ (c-$TiO_2$) are retained in the a-$TiO_2$ model.

In order to obtain the a-$TiO_2$ surface in the present study, a vacuum of about 20 Å was added in the z-direction of $2 \times 2 \times 4$ bulk supercell (96-atom model)[54]. The a-$TiO_{2-x}$ model was created by removing the surface O atoms coordinated to Ti having coordination numbers of four, five, and six, denoted by Ti-4c, Ti-5c, and Ti-6c, respectively (Fig. 4a), creating, in total, 10 a-$TiO_{2-x}$ model surfaces. For c-$TiO_2$, we investigated the most thermodynamically stable (101) surface of anatase with two models of crystalline $TiO_{2-x}$ (c-$TiO_{2-x}$) surface created by removing a two-fold coordinated O atom (denoted as O2c) and a three-fold coordinated O atom (denoted as O3c) (Supplementary Fig. 26). The anatase (101) surface is represented by a periodic supercell slab of 144 atoms and a vacuum region greater than 12 Å.

The plane wave pseudopotential approach, together with the Perdew-Burke-Ernzerhof[55] exchange-correlation functional and Vanderbilt ultrasoft pseudopotentials[56], was utilized throughout. The kinetic energy cutoffs of 544 and 5440 eV were used for the smooth part of the electronic wavefunctions and augmented electron density, respectively. The Quantum-ESPRESSO software package[57], was used to perform the calculations. All calculations were spin polarized. The structures were relaxed by using a conjugate gradient minimization algorithm until the magnitude of residual Hellman-Feynman force on each atom was less than $10^{-2}$ Ry/Bohr. In all electronic density of states (DOS) plots, a conventional Gaussian smearing of 0.007 Ry was utilized.

Appreciable underestimation of band gap and delocalization of d and f electrons are well-known limitations of DFT. Therefore, DFT with Hubbard energy correction (DFT + U) formalism was used in this study with U = 4.2 eV applied to Ti 3d electrons for analyzing electronic properties of crystalline and amorphous $TiO_2$. The value of U for Ti was chosen not solely on the basis of band gap but also depending on the property of interest[58], which in the current study is the catalytic behavior of $TiO_2$ that in-turn depends upon the position of band gap

states and their effect on the electronic structure. This value of U for Ti is consistent with theoretical investigations by ref. 59, who calculated it by fitting the peak positions for surface oxygen vacancies to experimental X-ray photoelectron spectroscopy data.

In order to understand the charge mobility (i.e., localization of energy states in a-$TiO_{2-x}$ and c-$TiO_{2-x}$), we further conducted Inverse Participation Ratio (IPR) analysis. IPR is capable of determining the charge localization of the tail states and the mobility band gap of materials[60]. The IPR of an orbital $\psi_n(\vec{r}_i)$, $I(\psi_n)$, is accordingly defined by

$$I(\psi_n) = N \frac{\sum_{i=1}^{N} |\psi_n(\vec{r}_i)|^4}{\left[\sum_{i=1}^{N} |\psi_n(\vec{r}_i)|^2\right]^2} \qquad (5)$$

where N is the number of volume elements in the cell and i is the index of the volume element. Ideally, a localized orbital means $I(\psi) = N$, whereas a delocalized orbital means $I(\psi) = 1$. The IPR can identify a level as belonging to the delocalized band, to the partially localized band tail, or to the highly localized band gap.

## Data availability

The data that support the plots within this paper and other finding of this study are available from the corresponding author upon reasonable request.

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

## Acknowledgements

G.A.O. acknowledges the financial support of the Ontario Ministry of Research and Innovation (MRI), the Ministry of Economic Development, Employment and Infrastructure (MEDI), the Ministry of the Environment and Climate Change's (MOECC) Best in Science (BIS) Award, Ontario Center of Excellence Solutions 2030 Challenge Fund, Ministry of Research Innovation and Science (MRIS) Low Carbon Innovation Fund (LCIF), Imperial Oil, the University of Toronto's Connaught Innovation Fund (CIF), Connaught Global Challenge (CGC) Fund, and the Natural Sciences and Engineering Research Council of Canada (NSERC). L.H. and X.H.Z. acknowledges the support from the National Natural Science Foundation of China (51920105005, 51821002, 21902113), 111 Project, and the Collaborative Innovation Center of Suzhou Nano Science & Technology. K.K.G. acknowledges Calcul Québec (www.calculquebec. ca) and Compute Canada (www.computecanada.ca) supercomputing facility; infrastructural support from Canada Foundation for Innovation, and the financial support from the Natural Sciences and Engineering Research Council of Canada (NSERC) Discovery grant program, (RGPIN-2020-05924) and the Canada Research Chairs Program. C.M. acknowledges financial support from the UofT Faculty of Arts & Science Postdoctoral Fellowship. P.N.D. acknowledges financial support from the NSERC PDF program. This research used resources of the Advanced Photon Source, a U.S. Department of Energy (DOE) Office of Science User Facility operated for the DOE Office of Science by Argonne National Laboratory under Contract no. DE-AC02-06CH11357. A. Tountas' help on ASPEN estimation is acknowledged by all authors.

## Author contributions

Z.L. and C.M. contributed equally to this work. G.A.O., X.Z., and Z.L. conceived and designed the experiments. Z.L. and Q.P. carried out the synthesis of the materials and catalytic testing. K.K.G. conducted and analyzed the DFT calculations. Z.L., C.M., M.X., J.W., F.M.A., and D.M.M. performed materials characterizations. Z.L., C.M., L.W., T.H., R.S., Q.G., and P.N.D. contributed to data analysis. G.A.O., X.Z., and K.K.G. supervised the project. G.A.O., X.Z., K.K.G., L.H., C.M., and Z.L. wrote the paper. P.N.D. edited the paper. All authors discussed the results and commented on the manuscript.

## Competing interests

The authors declare no competing interests.
