## [Peer Review File · Nature Communications]

Engineered Disorder in CO₂ PhotocatalysisREVIEWER COMMENTS

Reviewer #1 (Remarks to the Author):

This manuscript presents the beneficial synergy between structural disorder and surface frustrated Lewis pairs in photocatalytic CO₂ hydrogenation, with excellent CO production rate and TOF. It is helpful for designing SFLPs-base photocatalysts. However, This manuscript does not bring any new knowledge on materials property and therefore only contribution may be in novel preparation method. The mechanisms of CO₂ photocatalytic reactivity and core-shell heterostructure are not very clear; there are several statements not supported with data and even some results cannot distinguish between high precision data and errors. For these reasons, I suggest to reject this paper in the present form for publication in Nature Communications.

Detailed comments are as follows:

- (1) The authors proposed "UV can penetrate the a-TiO₂-x(OH)_y thick layer to reach and excite the c-TiO₂ component, generating electron-hole pairs", but from TDOS and PDOS results, the bandgap of amorphous a-TiO₂-x(OH)_y shell is smaller than crystalline c-TiO₂ core, why the photo-induced electron-hole pairs generate directly in shell layer? That is, why the UV can excite the a-TiO₂-x(OH)_y component?
- (2) Mulliken charge is coarse for calculating atomic charge, 0.01 e may be an error with this method. while the 0.002 Angstrom of O-H bond length difference is also maybe within the limit of error.
- (3) No band structure found in the whole manuscript and SI.
- (4) From Fig. 4b,c, in the energy range of -1 - 0 eV, there is a slight TDOS peak with high IPR for both a-TiO₂-x(OH)_y and TiO₂-x, but not appear in the TDOS /PDOS results in Supplementary Fig. 10.
- (5) Phonon calculations should be appended to provide the structural stabilities for both surface structures in DFT. Especially, the c-TiO₂ structure in Fig.5 seems to be a monoatomic layer. Generally, this is unreasonable for surface structure.
- (6) The authors presented the TOF of c-TiO₂@a-TiO₂-x(OH)_y was high and superb among known catalysts, but did not provide the comparison. This manuscript presents the beneficial synergy between structural disorder and surface frustrated Lewis pairs in photocatalytic CO₂ hydrogenation, with excellent CO production rate and TOF. It is helpful for designing SFLPs-base photocatalysts.
- (7) From the Fig.6a, increasing the component of SFLPs will notably improve the CO production rate, so what is the role for c-TiO₂ core?
- (8) The authors proposed the photo-electron will be trapped by an unsaturated Ti(III) and/or oxygen-vacancy sites in a-TiO₂-x(OH)_y, while photo-hole trapped by hydroxyl. Is there any result prove it, band-decomposed charge density? From the structure of SFLPs, the Ti(III) or oxygen-vacancy site is near the hydroxyl group, how to explain the charge carrier spatial separation? The carrier mobility or effective mass should be calculated to prove charge carrier transfer abilities.

Reviewer #2 (Remarks to the Author):

This manuscript created surface frustrated Lewis pairs (SFLPs) in an amorphous a-TiO₂-x(OH)_y surface. It discusses in detail the structure-activity relationship between the amorphous structure and the CO₂ reduction reaction. This work is meaningful for studying the influence of surface disorder structure on photocatalytic CO₂ reduction mechanism. It is a topic of interest to the researchers in the related areas, but the paper needs significant improvement before publication.

1. In the part of "Ti(III)-[O]-Ti(IV)-OH-related atomic/electronic structure", the author used solid-state 1H magic-angle spinning nuclear magnetic resonance (1H MAS NMR) to distinguish between SFLP [Ti(III)-[O]-Ti(IV)-OH] and conventional Bronsted acid [Ti(IV)-OH-Ti(IV)]. The c-TiO₂@a-TiO₂-x(OH)_y showed peaks at 5.2, 2.2, and 0.6 ppm, lower than those of c-TiO₂ with peaks at 5.8, 2.4, 1.6 ppm. What does the movement of the peak represent? It would be better if the authors discussed it in

more detail. Can the NMR peaks be assigned?

2. In Figure S7, weak peaks in the bridging region emerged on the c-TiO₂ sample, while large peak intensities with increased peak numbers were observed on the c-TiO₂@ a-TiO₂-x(OH)_y surface. Does the enhancement of the bridging OH feature mean the generation of Bronsted acid [Ti(IV)-OH-Ti(IV)]? In this case, SFLP [Ti(III)-[O]-Ti(IV)-OH] and conventional Bronsted acid [Ti(IV)-OH-Ti(IV)] are co-existed on the c-TiO₂ surface?

3. The author described the electron transfer process in the photocatalytic system. It mentioned that UV can penetrate the a-TiO₂-x(OH)_y thick layer to reach and excite the c-TiO₂ component under light irradiation. Can surface disordered structures with narrow gaps absorb the rest of visible light? What role does amorphous shell play in light absorption?

4. UV can penetrate the a-TiO₂-x(OH)_y thick layer to reach and excite the c-TiO₂ component under light irradiation. In the part of "Measurements of the gas-phase photocatalytic reduction CO₂", why 420 nm optical cut-off filter was used to remove UV light from the Xe source?

5. Figure 6a showed the more disordered structure, the stronger the photocatalytic performance. Is there an optimal value of reduction degree give the best photocatalytic performance?

6. The CO rate under about 700 nm wavelength light should be measured and shown in Figure S18. Does the author consider the corresponding apparent quantum efficiency(AQY)?

7. The [Ti(III)-[O]- Ti(IV)-OH] on the material surface boost photoreactivity towards H₂ heterolysis and CO₂ reduction. However, after the reaction is completed, can the bond be maintained for a long time and stably. We suggest the authors give the long-term test results.

8. The band structure can be changed by the introduction of an amorphous shell. However, the bandgap of the materials is not suitable for photocatalytic. Considering the previously reported excellent band gap of black titanium oxide, the advantage of this structure should be discussed.

9. Most of the article discusses the role of interface chemical bonds, while there is less discussion about amorphous shells. Is it possible to build similar chemical bonds at the interface of narrow band gap materials such as Ti₂-xO₂ to boost the reaction instead of having an amorphous structure?

10. The influence of the content of the amorphous structure or the [Ti(III)-[O]- Ti(IV)-OH] bonds should be explained. If there is optimal content, the paper should present the results.

Point-by-point response

Reviewer #1 (Remarks to the Author):

This manuscript presents the beneficial synergy between structural disorder and surface frustrated Lewis pairs in photocatalytic CO₂ hydrogenation, with excellent CO production rate and TOF. It is helpful for designing SFLPs-base photocatalysts. However, this manuscript does not bring any new knowledge on materials property and therefore only contribution may be in novel preparation method. The mechanisms of CO₂ photocatalytic reactivity and core-shell heterostructure are not very clear; there are several statements not supported with data and even some results cannot distinguish between high precision data and errors. For these reasons, I suggest to reject this paper in the present form for publication in Nature Communications.

Reply: We appreciate the referee's positive comments on our novel preparation method to obtain the black titania. We would like to emphasize that the obtained black titania showed superior performance towards photo(thermal) CO₂-to-CO catalysis, which can compete with reported photothermal catalysts consisting of metal oxides supported transitional metals (Supplementary Table 4). Furthermore, we provide the first molecular-level understanding—the SFLP—to the amorphous black titania during the photo(thermo)catalysis, which offers an understanding advance in the field. During this revision, we added additional experimental results (characterizations and control activity tests), calculations on light penetration depth and TDOS, explanatory sentences and corrected typos to resolve concerns related to the mechanism. **Actions taken are indicated in blue** while **explanatory sentences are shown in red**.

Detailed comments are as follows:

(1) The authors proposed "UV can penetrate the a-TiO_{2-x}(OH)_y thick layer to reach and excite the c-TiO₂ component, generating electron-hole pairs", but from TDOS and PDOS results, the bandgap of amorphous a-TiO_{2-x}(OH)_y shell is smaller than crystalline c-TiO₂ core, why the photo-induced electron-hole pairs generate directly in shell layer? That is, why the UV can excite the a-TiO_{2-x}(OH)_y component?

Reply: Thanks for the referee's comment, however, we cannot see any conflicts on this point. Theoretically, the smaller the bandgap of a semiconductor, the easier it is to be excited by light. In our case, the UV light can excite both the core c-TiO₂ and the shell a-TiO_{2-x}(OH)_y because the photon energy of the UV light ($\lambda < 347$ nm or $E > 3.57$ eV) is larger than the bandgap energy of c-TiO₂ ($E_g = 3.57$ eV) or a-TiO_{2-x}(OH)_y ($E_g = 3.38$ eV; Supplementary Fig. 9). Accordingly, the visible and IR light can also excite electrons on the defect- and OH-related energy levels of the shell a-TiO_{2-x}(OH)_y, which is shown in Fig. 4d (below picture).

To further support above claims, the penetration depth (D_p) of UV light can be calculated for TiO_2 : The $D_p = 1/\alpha$, where α is the light absorption coefficient that can be calculated using the formula of $\alpha = 4\pi k/\lambda$, where k is the extinction coefficient that is documented for white TiO_2 ¹, λ is the wavelength of the incident light. Based on above calculation, the wavelength-dependent D_p is listed below for white TiO_2 . According to the reflection spectra (Fig. 4a), $c\text{-TiO}_2$ and $a\text{-TiO}_{2-x}(\text{OH})_y$ demonstrates close UV-absorbing capacities, thus hinting the similar penetration depth in UV region. Given the D_p is larger than 10 nm, the UV light will be partially absorbed by the 2–6 nm $a\text{-TiO}_{2-x}(\text{OH})_y$ shell and then reach the core $c\text{-TiO}_2$ in $c\text{-TiO}_2@a\text{-TiO}_{2-x}(\text{OH})_y$ (Figure R1 and R2). Therefore, both the core $c\text{-TiO}_2$ and shell $a\text{-TiO}_{2-x}(\text{OH})_y$ will be excited by the UV light and then yield photo electrons and holes. We added the calculation results, schematic and corresponding discussion on page 10-11, line 273-290 and in Supplementary Fig. 11 and Fig. 12.

Figure R1. Wavelength-dependent light absorption coefficient and penetration depth of the TiO_2 . The optical constants of anatase TiO_2 are obtained from Ref¹ by the average of parallel and perpendicular optical constants. The TiO_2 is transparent to visible light and infrared light with energy smaller than its bandgap energy, and thus the penetration depth can only be calculated within the UV region.

Figure R2. Schematic of light penetration in stacked $c\text{-TiO}_2@a\text{-TiO}_{2-x}(\text{OH})_y$ catalyst.

Reference

1. Palik ED. Handbook of optical constants of solids. Elsevier Science, 1998.

(2) Mulliken charge is coarse for calculating atomic charge, 0.01 e may be an error with this method. while the 0.002 Angstrom of O-H bond length difference is also maybe within the limit of error.

Reply: Agreed. We have deleted the coarse-accuracy (for screening purpose only as stated in the manuscript) Mulliken charge and geometric calculation in Fig. 3d to avoid any misunderstandings. Instead, we add a theoretic schematic to the SFLP model and a detailed analysis of the ^1H -NMR result to identify the SFLP protons, and focus on the high-accuracy Bader charge calculation (Supplementary Table 2) based on the amorphous structure to understand the charge distribution in the SFLP on page 13, line 337-340.

“Theoretically, the terminal hydroxyl Ti(IV)-OH in the SFLP possesses higher degree of freedom compared to that of the bridging hydroxyl (Fig. 3d). Thus, their charge density and molecular vibrations should be distinct, which can be monitored through solid-state ^1H magic-angle spinning nuclear magnetic resonance (^1H MAS NMR) and attenuated total reflection Fourier transform infrared (ATR-FTIR), respectively. The ^1H MAS NMR spectrum of $c\text{-TiO}_2$ showed peaks at 5.8, 1.6 and -1.65 ppm. The peak around 5.8 ppm is typically ascribed to absorbed water¹, and the peak at 2.2 ppm on neat titanium dioxide is assigned to terminal hydroxyl group. Both water and hydroxyl group are naturally existing species when a metal oxide sample is exposed to air. The peak at -1.65 ppm appeared to be a background signal as similar signal emerged on the ^1H MAS NMR spectra of $c\text{-TiO}_2@a\text{-TiO}_{2-x}(\text{OH})_y$ as well. Interestingly, the water and hydroxyl peaks of $c\text{-TiO}_2@a\text{-TiO}_{2-x}(\text{OH})_y$ shifted to 5.2 ppm and 1.7 ppm, respectively,

and two additional peaks (2.2 and 0.6 ppm)^{2,3} emerged around the hydroxyl peak. Since the chemical shift of ¹H-NMR reflects the shielding effect, compared to c-TiO₂, the lower chemical shift of water on c-TiO₂@a-TiO_{2-x}(OH)_y indicated the possible electron donation from oxygen vacancies and/or Ti(III) in a-TiO_{2-x}(OH)_y, while the higher chemical shift of hydroxyl (1.7 and 2.2 ppm) suggested the proton of less electron density, which agreed well the Lewis base -OH in the SFLP model. The peak at 0.6 ppm can be assigned to several species, including terminal hydroxyl group and proton bonded to oxygen vacancy, which are closely related to the amorphous a-TiO_{2-x}(OH)_y shell (Fig. 3e)” on page 7-8, line 186-206.

Reference

1. Nosaka AY, Fujiwara T, Yagi H, Akutsu H, Nosaka Y. Characteristics of water adsorbed on TiO₂ photocatalytic systems with increasing temperature as studied by solid-state ¹H NMR spectroscopy. *J. Phys. Chem. B* 2004, 108(26): 9121-9125.
2. Li G, et al. Ionothermal synthesis of black Ti³⁺-doped single-crystal TiO₂ as an active photocatalyst for pollutant degradation and H₂ generation. *J. Mater. Chem. A* 2015, 3(7): 3748-3756.
3. Crocker M, et al. ¹H NMR spectroscopy of titania. Chemical shift assignments for hydroxy groups in crystalline and amorphous forms of TiO₂. *J. Chem. Soc., Faraday Trans.* 1996, 92(15): 2791-2798.

(3) No band structure found in the whole manuscript and SI.

Reply: Thanks for your comment, however, the experimental valence/conduction band positions and bandgap energies are already provided for both c-TiO₂ and c-TiO₂@a-TiO_{2-x}(OH)_y (Fig. 4), which are determined using a combination of UV-VIS-NIR, Tauc plot (Fig. 4a and Supplementary Fig. 8) and UPS spectra (Supplementary Fig. 9).

Typically, theoretic band structure is calculated and plotted against high symmetry points of a crystal in DFT. However, in an amorphous solid devoid of symmetry, the physic meaning of band structure calculation is ambiguous, and thus it is not usually possible to determine a precise dispersion relation. Therefore, no theoretic band structure result is given but instead the TDOS (Fig. 4b, c and Supplementary Fig.10) to help understand the experimental results.

(4) From Fig. 4b,c, in the energy range of -1 - 0 eV, there is a slight TDOS peak with high IPR for both a-TiO_{2-x}(OH)_y and TiO_{2-x}, but not appear in the TDOS /PDOS results in Supplementary Fig. 10.

Reply: We appreciate the referee’s vital comment and apologize for the oversight in the Fig. 4b. We double-checked the result and found that the original Fig. 4b was the result for amorphous TiO_{2-x} rather than the labeled amorphous TiO_{2-x}(OH)_y. Therefore, the TDOS in Fig. 4b was same to that in Supplementary Fig.10 (amorphous TiO_{2-x}). We have revised Fig. 4b by a new calculation based on the amorphous TiO_{2-x}(OH)_y, and

the new result (Figure. R3) has been supplemented to replace the previous one. Now the TDOS in Fig. 4b agrees with that in Supplementary Fig.10a ($\text{TiO}_{2-x}(\text{OH})_y$), and the major claim of “SFLP-facilitated charge localization” related to Fig. 4b remained valid. Fig. 4c represents the **crystalline** TiO_{2-x} which does not present in the Supplementary Fig.10, so there is no conflict.

Figure R3. The total density of states (black line, right axis) and the corresponding values of the inverse participation ratio (IPR) (blue dots, left axis) for the a- $\text{TiO}_{2-x}(\text{OH})_y$ surface.

(5) Phonon calculations should be appended to provide the structural stabilities for both surface structures in DFT. Especially, the c- TiO_2 structure in Fig.5 seems to be a monoatomic layer. Generally, this is unreasonable for surface structure.

Reply: We appreciate the referee’s helpful comment for pointing out the possible misunderstanding in the DFT model. The c- TiO_2 model used in this study is not a monoatomic layer but a slab structure with multiple atomic layers and a vacuum slab. The previous monoatomic layer-like Fig. 5e is due to the periodicity of crystalline TiO_{2-x} when observed from the top-view. We have made this point clear by tuning the 3-dimensional representation of c- TiO_{2-x} structure into a side-view mode (Figure R4) in Fig. 5e in manuscript and Supplementary Fig. 14 in SI. Also, the top-view picture of the c- TiO_2 was supplemented in Supplementary Fig. 25.

If a phonon frequency is positive, that means a positive curvature of the potential energy surface. So the energy increases quadratically if atoms are placed in the directions given by the associated eigenvector. Otherwise, the imaginary (or “negative”) frequency indicates an energy decrease, or known as an “unstable” structure for a DFT model that does reach local free energy minimum. To this regard, phonon calculations are usually performed in transient-state search to verify the “stability” of intermediate models where conventional geometric optimization to search for the energy minimum is not applied.

However, in our steady-state, conventional geometric optimization, the structural stabilities of our crystalline and amorphous structures are promised by performing the standard DFT optimization via conjugate gradient minimization algorithm until the magnitude of residual Hellman-Feynman force on each atom was less than 10^{-2} Ry/Bohr. This geometric optimization is superior to phonon calculation to this case. Furthermore, our amorphous TiO₂ models and methodology to model amorphous TiO₂ are also extensively validated in our previous works.^{1, 2, 3, 4, 5, 6}

Figure R4. Interaction of H₂ on anatase (101) surface with oxygen vacancy (c-TiO_{2-x}) at O-2c site (left: side-view) and O3c site (middle; side-view), and the top-view of the c-TiO₂ lattice (right).

Reference

1. Ghuman KK, Singh CV. Self-trapped charge carriers in defected amorphous TiO₂. *J. Phys. Chem. C* 2016, **120**(49): 27910-27916.
2. Ghuman KK. Mechanistic insights into water adsorption and dissociation on amorphous TiO₂-based catalysts. *Science and Technology of Advanced Materials* 2018, **19**(1): 44-52.
3. Ghuman KK, Singh CV. Effect of doping on electronic structure and photocatalytic behavior of amorphous TiO₂. *J. Phys.: Condens. Matter* 2013, **25**(47): 475501.
4. Kaur K, Singh CV. Amorphous TiO₂ as a photocatalyst for hydrogen production: A DFT study of structural and electronic properties. *Energy Procedia* 2012, **29**: 291-299.
5. Ghuman KK, Goyal N, Prakash S. Vibrational density of states of TiO₂ nanoparticles. *J. Non-Cryst. Solids* 2013, **373-374**: 28-33.
6. Kaur K, Prakash S, Goyal N, Singh R, Entel P. Structure factor of amorphous TiO₂ nanoparticle; Molecular Dynamics Study. *J. Non-Cryst. Solids* 2011, **357**(19): 3399-3404.

(6) The authors presented the TOF of c-TiO₂@a-TiO_{2-x}(OH)_y was high and superb among

known catalysts, but did not provide the comparison. This manuscript presents the beneficial synergy between structural disorder and surface frustrated Lewis pairs in photocatalytic CO₂ hydrogenation, with excellent CO production rate and TOF. It is helpful for designing SFLPs-base photocatalysts.

Reply: Thanks for the referee's comment, however, the comparison for CO₂ photocatalysis is already shown in Supplementary Table 3. The activity of our catalyst, c-TiO₂@a-TiO_{2-x}(OH)_y, is comparable to those achieved over supported metal photothermocatalysts in CO₂-to-CO.

(7) From the Fig.6a, increasing the component of SFLPs will notably improve the CO production rate, so what is the role for c-TiO₂ core?

Reply: Thanks for reviewer's vital comment on clarifying the mechanism. The SFLP is a concomitant in the Na-treatment of the c-TiO₂ surface to generate the a-TiO_{2-x}(OH)_y shell. One feature of a-TiO_{2-x}(OH)_y is processing high concentration of oxygen vacancy. It is well known that oxygen vacancies can act as electron traps, and thus surface oxygen vacancies can facilitate surface reaction due to elongated lifetime of charge carriers, but dense bulk oxygen vacancies severely hamper the charge separation and transportation to reach the surface^{1,2}.

The efficiency of a photocatalytic reaction is determined by the efficiencies of light absorption, charge separation and finally the surface reaction. The surface reactivity and light absorption of a-TiO_{2-x}(OH)_y catalyst should be similar to that of c-TiO₂@a-TiO_{2-x}(OH)_y when stacked together in the powder form, and the light penetration depth of the former should be smaller than the latter, as shown in Figure R5. Due to limited charge separation of a-TiO_{2-x}(OH)_y compared with the c-TiO₂@a-TiO_{2-x}(OH)_y, the activity of a-TiO_{2-x}(OH)_y should be lower than that of the c-TiO₂@a-TiO_{2-x}(OH)_y due to smaller surface area induced by the light penetration depth (Figure R6). The whole picture of the mechanism is shown in Fig. 1 and Supplementary Fig. 12.

Figure R5. Schematic of the light penetration depth and effective photochemistry surface area for $c\text{-TiO}_2@a\text{-TiO}_{2-x}(\text{OH})_y$ and pure $a\text{-TiO}_{2-x}(\text{OH})_y$ in stacked powder form. The total light absorption of the two samples in powder form should be the same which equals to $(1-R)$, where R is the reflectance by the surface, but the Vis-IR light penetration depth of $c\text{-TiO}_2@a\text{-TiO}_{2-x}(\text{OH})_y$ should be larger than that of $a\text{-TiO}_{2-x}(\text{OH})_y$ due to the Vis-IR transparent $c\text{-TiO}_2$ core. Thus, the light accessible surface area of the former should be larger than the latter.

Figure R6. a. EPR spectra of $c\text{-TiO}_2@a\text{-TiO}_{2-x}(\text{OH})_y$. b. CO production rates of $c\text{-TiO}_2@a\text{-TiO}_{2-x}(\text{OH})_y$ with different oxygen vacancy concentrations, under full-spectrum Xe light (4.0 W cm^{-2}).

Reference

1. Kong M, et al. Tuning the relative concentration ratio of bulk defects to surface defects in TiO_2 nanocrystals leads to high photocatalytic efficiency. *J. Am. Chem.*

Soc. 2011, 133(41): 16414-16417.

2. Mao C, et al. Beyond the thermal equilibrium limit of ammonia synthesis with dual temperature zone catalyst powered by solar light. Chem 2019, 5(10): 2702-2717.

(8) The authors proposed the photo-electron will be trapped by an unsaturated Ti(III) and/or oxygen-vacancy sites in a-TiO_{2-x}(OH)_y, while photo-hole trapped by hydroxyl. Is there any result prove it, band-decomposed charge density? From the structure of SFLPs, the Ti(III) or oxygen-vacancy site is near the hydroxyl group, how to explain the charge carrier spatial separation? The carrier mobility or effective mass should be calculated to prove charge carrier transfer abilities.

Reply: Thanks for reviewer's insightful comment on the charge transportation. The claim related to the feat of charge carriers is made based on DFT calculations and experimental results: **Theoretically, O vacancy (Ti(III)) and hydroxyl (OH) results in mid band gap states below the CB minima and above the VB maxima, respectively. The mid-gap states of OH(-I) are induced by O 2p orbitals and Ti(III) and [O]_v mid-gap states originate from Ti 3d orbitals (Fig. 4d and Supplementary Fig. 10a). These OH(-I), Ti(III) and [O]_v states at the band edges are highly localized (with low mobility) as indicated by the IPR analysis (Fig. 4b), which we expect to act as the trapping centers for electrons and holes.^{1, 2} These points are further supported by UPS and PL results. To amplify, the UPS result evidenced VB upshift and CB downshift after introducing the a-TiO_{2-x}(OH)_y on the surface of c-TiO₂. Upon light excitation, the holes will be generated on lattice oxygen and OH sites which finally relaxes to OH(-I) sites, while the electrons excited to Ti(IV), Ti(III) and [O]_v sites will relax to [O]_v sites. Given the relaxation of holes and electrons involves intra- and/or inter-band transitions, the lifetimes of holes/electrons will be prolonged in c-TiO₂@a-TiO_{2-x}(OH)_y compared to those in pure c-TiO₂, which has been evidenced by the elongated lifetime derived from the PL decay curve in Fig. 4e.**

Based on above results, the spatial electron transfer is either from O(-II) to Ti(III)/Ti(IV) or from OH(-I) to Ti(IV), [O]_v and then Ti(III), as supplemented in revised Fig. 1a and 1e (below picture, the orange arrows in Fig. 1b and 1e indicate the spatial electron transfer while the black arrows in Fig. 1a indicate the charge carrier transfer in terms of energy band levels).

The effective mass is associated with the electronic band curvature along a specific direction, which is proportional to the second derivative of the energy band with respect to the wave vector k. Given the DFT calculation of effective mass of charge carriers are typically realized through mathematic calculation on top of band structure dispersion function, the amorphous structure devoid of high symmetry cannot get a band structure dispersion, nor to perform such a kind of calculation. Instead, the IPR analysis is provided to understand the carrier mobility (Fig. 4b).

Reference:

- Hayes W, Jenkin TJL. Charge-trapping properties of germanium in crystalline quartz. *Journal of Physics C: Solid State Physics* 1986, 19(31): 6211-6219.
- Ghuman KK. Mechanistic insights into water adsorption and dissociation on amorphous TiO₂-based catalysts. *Science and Technology of Advanced Materials* 2018, 19(1): 44-52.

Reviewer #2 (Remarks to the Author):

This manuscript created surface frustrated Lewis pairs (SFLPs) in an amorphous a-TiO_{2-x}(OH)_y surface. It discusses in detail the structure-activity relationship between the amorphous structure and the CO₂ reduction reaction. This work is meaningful for studying the influence of surface disorder structure on photocatalytic CO₂ reduction mechanism. It is a topic of interest to the researchers in the related areas, but the paper needs significant improvement before publication.

Reply: We appreciate the referee's positive comments on the novelty of our work. During this revision, we added additional experimental results (control activity tests), calculations (light penetration depth) and explanatory sentences to resolve concerns

related to the mechanism. Actions taken are indicated in blue while explanatory sentences are shown in red.

1. In the part of "Ti(III)-[O]-Ti(IV)-OH-related atomic/electronic structure", the author used solid-state ^1H magic-angle spinning nuclear magnetic resonance (^1H MAS NMR) to distinguish between SFLP [Ti(III)-[O]-Ti(IV)-OH] and conventional Bronsted acid [Ti(IV)-OH-Ti(IV)]. The $c\text{-TiO}_2@a\text{-TiO}_{2-x}(\text{OH})_y$ showed peaks at 5.2, 2.2, and 0.6 ppm, lower than those of $c\text{-TiO}_2$ with peaks at 5.8, 2.4, 1.6 ppm. What does the movement of the peak represent? It would be better if the authors discussed it in more detail. Can the NMR peaks be assigned?

Reply: Agreed and we have added detailed ^1H -NMR analysis on page 7-8, line 191-206. "The ^1H MAS NMR spectrum of $c\text{-TiO}_2$ showed peaks at 5.8, 1.6 and -1.65 ppm. The peak around 5.8 ppm is typically ascribed to absorbed water, and the peak at 2.2 ppm on neat titanium dioxide is assigned to hydroxyl group. Both water and hydroxyl group are naturally occurring species when a metal oxide sample is exposed to air. The peak at -1.65 ppm appeared to be a background signal as similar signal emerged on the ^1H MAS NMR spectra of $c\text{-TiO}_2@a\text{-TiO}_{2-x}(\text{OH})_y$ as well. Interestingly, the water and hydroxyl peaks of $c\text{-TiO}_2@a\text{-TiO}_{2-x}(\text{OH})_y$ shifted to 5.2 ppm and 1.7 ppm, respectively, and two additional peaks (2.2 and 0.6 ppm)^{1, 2, 3} emerged around the hydroxyl peak. Since the chemical shift of ^1H -NMR reflects the shielding effect, compared to $c\text{-TiO}_2$, the lower chemical shift of water on $c\text{-TiO}_2@a\text{-TiO}_{2-x}(\text{OH})_y$ indicated the possible electron donation from oxygen vacancies and/or Ti(III) in $a\text{-TiO}_{2-x}(\text{OH})_y$, while the higher chemical shift of hydroxyl (1.7 and 2.2 ppm) suggested the proton of less electron density, which agreed well the Lewis base -OH in the SFLP model. The peak at 0.6 ppm can be assigned to several species, including terminal hydroxyl group and proton bonded to oxygen vacancy, which are closely related to the amorphous $a\text{-TiO}_{2-x}(\text{OH})_y$ shell (Fig. 3e)."

Reference

1. Nosaka AY, Fujiwara T, Yagi H, Akutsu H, Nosaka Y. Characteristics of water adsorbed on TiO_2 photocatalytic systems with increasing temperature as studied by solid-state ^1H NMR spectroscopy. *J. Phys. Chem. B* 2004, 108(26): 9121-9125.
2. Li G, et al. Ionothermal synthesis of black Ti^{3+} -doped single-crystal TiO_2 as an active photocatalyst for pollutant degradation and H_2 generation. *J. Mater. Chem. A* 2015, 3(7): 3748-3756.
3. Crocker M, et al. ^1H NMR spectroscopy of titania. Chemical shift assignments for hydroxy groups in crystalline and amorphous forms of TiO_2 . *J. Chem. Soc., Faraday Trans.* 1996, 92(15): 2791-2798.

2. In Figure S7, weak peaks in the bridging region emerged on the $c\text{-TiO}_2$ sample, while large peak intensities with increased peak numbers were observed on the $c\text{-TiO}_2@a\text{-TiO}_{2-x}(\text{OH})_y$ surface. Does the enhancement of the bridging OH feature mean the generation of Bronsted

acid [Ti(IV)-OH-Ti(IV)]? In this case, SFLP [Ti(III)-[O]-Ti(IV)-OH] and conventional Bronsted acid [Ti(IV)-OH-Ti(IV)] are co-existed on the c-TiO₂ surface?

Reply: We appreciate the reviewer's comments and agree with the co-existence of bridging and terminal OH in c-TiO₂@a-TiO_{2-x}(OH)_y. The generation of bridging OH is found to be oxygen vacancy-dependent H₂O heterolysis, which is well-documented in literatures^{1,2}. The process can be summarized as $\text{Ti-[O]}_v\text{-Ti-O} + \text{H}_2\text{O} = \text{Ti-OH-Ti-OH}$. Given the [O]_v-rich nature of a-TiO_{2-x}(OH)_y, the bridging OH generation is possible during the H₂O washing or atmospheric moisture adsorption, besides the terminal OH generation. However, the bridging OH (pure c-TiO₂) is inactive towards H₂ splitting and less active for CO₂ reduction compared to SFLP (c-TiO₂@a-TiO_{2-x}(OH)_y).

Reference

1. Petrik NG, Kimmel GA. Reaction Kinetics of Water Molecules with Oxygen Vacancies on Rutile TiO₂(110). *J. Phys. Chem. C* 2015, **119**(40): 23059-23067.
2. Zhang Z, Bondarchuk O, Kay BD, White JM, Dohnálek Z. Imaging Water Dissociation on TiO₂(110): Evidence for Inequivalent Geminate OH Groups. *J. Phys. Chem. B* 2006, **110**(43): 21840-21845.

3. The author described the electron transfer process in the photocatalytic system. It mentioned that UV can penetrate the a-TiO_{2-x}(OH)_y thick layer to reach and excite the c-TiO₂ component under light irradiation. Can surface disordered structures with narrow gaps absorb the rest of visible light? What role does amorphous shell play in light absorption?

Reply: We appreciate the referee's vital comments on the mechanism. The light absorption of a material is determined by its extinction coefficient and light penetration depth. The penetration depth (D_p) of UV light for a-TiO_{2-x}(OH)_y is larger than 10 nm, so the UV light can penetrate the 3-6 nm shell to reach and excite the core TiO₂. To amplify, the $D_p = 1/\alpha$, where α is the light absorption coefficient. α can be calculated from using the formula of $\alpha = 4\pi k/\lambda$, where k is the extinction coefficient that is well documented for white TiO₂,¹ λ is the wavelength of incident light. Based on above calculation, the wavelength-dependent D_p is listed below for white TiO₂ (Figure R1). According to the reflection spectra of black titania (Fig. 4a), its UV light absorption is close to the white TiO₂, thus the similar penetration depth of several hundred nanometers in UV region.

Figure R1. Wavelength-dependent light absorption coefficient and penetration depth of TiO₂. The optical constants of anatase TiO₂ from 300 to 2500 nm are obtained from Ref¹ by the average of parallel and perpendicular optical constants. The TiO₂ is transparent to visible light and infrared light with wavelengths above its bandgap energy.

According to accessible absorption/refraction index data in black titania², the visible light penetration depth of black titania is within several micrometers, so the visible light cannot be fully absorbed by a 3-6 nm surface a-TiO_{2-x}(OH)_y layer but will be absorbed by the subsurface catalyst particles with a stacking thickness reaching several micrometers (Figure R2). We also added this part and the corresponding discussion on page 10, line 273-286 in Supplementary Fig. 12.

Figure R2. Schematic of the light penetration depth and effective photochemistry surface area for $c\text{-TiO}_2@a\text{-TiO}_{2-x}(\text{OH})_y$ and pure $a\text{-TiO}_{2-x}(\text{OH})_y$ in the stacked powder form. The total light absorption between the two samples should be the same which equals to $(1-R)$, where R is the reflectance by the surface, but the Vis-IR light penetration depth of $c\text{-TiO}_2@a\text{-TiO}_{2-x}(\text{OH})_y$ should be larger than that of $a\text{-TiO}_{2-x}(\text{OH})_y$ due to the Vis-IR transparent $c\text{-TiO}_2$ core. Thus, the light accessible surface area of the former should be larger than the latter.

The role of amorphous shell in $c\text{-TiO}_2@a\text{-TiO}_{2-x}(\text{OH})_y$ can be clarified by comparison with pure $c\text{-TiO}_2$ or pure $a\text{-TiO}_{2-x}(\text{OH})_y$. Pure TiO_2 can only absorb UV light and is transparent to visible and infrared light, while pure $\text{TiO}_{2-x}(\text{OH})_y$ can absorb UV-Vis-IR light. In terms of converting the absorbed photons into charge carriers to drive chemical reaction, the pure TiO_2 is superior in e^-h^+ separation and subsequent transfer from bulk to surface compared to that in the $\text{TiO}_{2-x}(\text{OH})_y$, since the bulk oxygen vacancies in the latter would act as the deep trap for electrons^{3, 4, 5}.

The $c\text{-TiO}_2@a\text{-TiO}_{2-x}(\text{OH})_y$, on the other hand, can make the best use of the full-spectrum light-absorption capability of $\text{TiO}_{2-x}(\text{OH})_y$ and the charge-separation capability of TiO_2 by forming a core-shell hybrid structure. This claim is further supported by a non-stoichiometry-dependent activity test in Reply to Q5, where the optimal x value of $c\text{-TiO}_2@a\text{-TiO}_{2-x}(\text{OH})_y$ is $0.0015 < x < 0.0031$ and further increasing the x value of the catalyst results in activity decrease.

Reference

1. Palik ED. *Handbook of optical constants of solids*. Elsevier Science, 1998.
2. Mao C, et al. Beyond the thermal equilibrium limit of ammonia synthesis with dual temperature zone catalyst powered by solar light. *Chem* 2019, 5(10): 2702-2717.
3. Wang B, Shen S, Mao SS. Black TiO_2 for solar hydrogen conversion. *Journal of Materiomics* 2017, 3(2): 96-111.
4. Naldoni A, et al. Photocatalysis with reduced TiO_2 : From Black TiO_2 to cocatalyst-free hydrogen production. *ACS Catal.* 2019, 9(1): 345-364.
5. Kong M, et al. Tuning the relative concentration ratio of bulk defects to surface defects in TiO_2 nanocrystals leads to high photocatalytic efficiency. *J. Am. Chem. Soc.* 2011, 133(41): 16414-16417.

4. UV can penetrate the $a\text{-TiO}_{2-x}(\text{OH})_y$ thick layer to reach and excite the $c\text{-TiO}_2$ component under light irradiation. In the part of "Measurements of the gas-phase photocatalytic reduction CO_2 ", why 420 nm optical cut-off filter was used to remove UV light from the Xe source?

Reply: We appreciate the referee's vital comment. The core $c\text{-TiO}_2$ can only absorb UV light, while the absorbance of $c\text{-TiO}_2@a\text{-TiO}_{2-x}(\text{OH})_y$ expands to visible and IR light (Figure 4a). 420 nm cut-off test helps to clarify the contribution of UV portion in the full Xe lamp spectrum, and demonstrates the visible and IR light utilization of the $a\text{-TiO}_{2-x}(\text{OH})_y$ shell. We have added corresponding discussion in page 15, line 413-420.

5. Figure 6a showed the more disordered structure, the stronger the photocatalytic performance. Is there an optimal value of reduction degree give the best photocatalytic performance?

Reply: We appreciate the referee's insightful comment on the control test. As shown in Figure R3b, the CO production firstly increased with non-stoichiometry of $c\text{-TiO}_2@a\text{-TiO}_{2-x}(\text{OH})_y$ samples to reach the champion activity of $5.3 \text{ mmol g}_{\text{cat}}^{-1} \text{ h}^{-1}$ when $0.0015 < x < 0.0031$. Further increasing the non-stoichiometry to $0.0021 < x < 0.0039$ (Figure R3a) resulted in activity drop to back to $4.7 \text{ mmol g}_{\text{cat}}^{-1} \text{ h}^{-1}$. This result indicates that it is necessary to engineer the percentage of amorphous shell in $c\text{-TiO}_2@a\text{-TiO}_{2-x}(\text{OH})_y$ to make a trade-off between the concentration of surface-active sites and the separation of charge carriers. We also substituted Fig. 6a in the manuscript and added the corresponding description on page 15, line 397-402 in manuscript.

Figure R3. a. EPR spectra of $c\text{-TiO}_2@a\text{-TiO}_{2-x}(\text{OH})_y$. b. CO production rates of $c\text{-TiO}_2@a\text{-TiO}_{2-x}(\text{OH})_y$ with different oxygen vacancy concentrations, under full-spectrum Xe light (4.0 W cm^{-2}).

6. The CO rate under about 700 nm wavelength light should be measured and shown in Figure S18. Does the author consider the corresponding apparent quantum efficiency(AQY)?

Reply: We appreciate the reviewer's comments and apologize that our LED sources don't cover $> 700 \text{ nm}$ wavelength. Instead, we supplemented the spectra of our red, green and blue LED (Figure R4) and calculated the AQY.

Figure R4. The spectra of LED light source.

The apparent quantum yield (AQY) was calculated according to the following equation.

$$AQY = \frac{N(\text{electrons})}{N(\text{photons})} * 100\%$$

where N(electrons) and N(photons) represent the number of reacted electrons and the number of incident photons, respectively. According to the chemical equation ($\text{CO}_2 + \text{H}_2 \rightarrow \text{CO} + \text{H}_2\text{O}$), $N(\text{electrons}) = N(\text{CO}) = M(\text{CO})N_A$, where N(CO), M(CO) and N_A represent the number of produced CO molecules, the number of moles of CO and Avogadro's constant, respectively.

In this study, N(photon), is estimated from the light intensity dispersion of the Xe lamp and the UV-vis-NIR absorption spectra.

$$N_{\text{photon}} = \int_{300\text{nm}}^{2400\text{nm}} \frac{\text{Light intensity} * I\% * A\% * \text{illumination area} * \text{time}}{\text{Average single photo energy} * N_A}$$

Where the light intensity is 4 W, illumination area is 1 cm², I% is the percentage of the Xe light intensity at certain wavelength (Figure R5), A% is the light harvesting efficiency at certain wavelength according to the absorption spectra (Fig.4a in manuscript), time is 3600s. The average single photon energy (E_{photon}) is figured out using the equation: $E\lambda = hc/\lambda$, where h is the Planck constant, c indicates speed of light, and λ is the wavelength.

Thus, we can figure out that AQY for c-TiO₂@a-TiO_{2-x}(OH)_y is 0.09%. The AQY was add in manuscript on page 14, line 375-379 and the calculation was added in supplementary note.

Figure R5. The spectra of 300W Xe lamp.

7. The [Ti(III)-[O]- Ti(IV)-OH] on the material surface boost photoreactivity towards H_2 heterolysis and CO_2 reduction. However, after the reaction is completed, can the bond be maintained for a long time and stably. We suggest the authors give the long-term test results.

Reply: Appreciate the reviewer's comments. Long-term stability (48h) was shown in Supplementary Fig.16 and CO production kept steady. Theoretically, the surface OH can be recycled through H_2 splitting and oxygen vacancy can maintain via H_2 reduction. O1s XPS spectra for the catalyst before and after reaction evidenced the population of both [O]v and hydroxyl group only decreased slightly (Supplementary Fig.18), which is in good agreement of the stability test in Supplementary Fig. 16.

8. The band structure can be changed by the introduction of an amorphous shell. However, the bandgap of the materials is not suitable for photocatalytic. Considering the previously reported excellent band gap of black titanium oxide, the advantage of this structure should be discussed.

Reply: Agreed. The optical properties of semiconductor nanomaterials strongly depend on their band gaps. TiO_2 can be used for photocatalytic CO_2 reduction because of its suitable valence band (VB) and conduction band (CB) levels.^{1,2} Narrowing the bandgap of the TiO_2 will reduce its redox capability to exchange for the enhancement in light absorption. To this regard, previous black titanium oxide with reduced band gap³ should be excellent only for water splitting, instead of the CO_2 catalysis that requires high reduction potential.

The advantage of our crystalline core-amorphous shell is 1) increasing the light transportation path, 2) and thus increasing the effective surface area for photochemistry, 3) inheriting high reduction potential from TiO_2 , 4) combining the advantages of both TiO_2 and $a-TiO_{2-x}(OH)_y$ in charge separation and light absorption, respectively. These

merits promote the catalytic efficiency towards photo(thermal)catalytic CO₂ reduction, as evidenced by the activity test. Details can be found in Reply to Q3.

Reference:

1. Rawool SA, Yadav KK, Polshettiwar V. Defective TiO₂ for photocatalytic CO₂ conversion to fuels and chemicals. *Chem. Sci.* 2021, **12**(12): 4267-4299.
2. Habisreutinger SN, Schmidt-Mende L, Stolarczyk JK. Photocatalytic Reduction of CO₂ on TiO₂ and Other Semiconductors. *Angew. Chem. Int. Ed.* 2013, **52**(29): 7372-7408.
3. Chen X, Liu L, Yu PY, Mao SS. Increasing solar absorption for photocatalysis with black hydrogenated titanium dioxide nanocrystals. *Science* 2011, **331**(6018): 746-750.

9. Most of the article discusses the role of interface chemical bonds, while there is less discussion about amorphous shells. Is it possible to build similar chemical bonds at the interface of narrow band gap materials such as Ti_{2-x}O₂ to boost the reaction instead of having an amorphous structure?

Reply: Appreciate the reviewer's comments and we agree it is theoretically possible to craft SFLPs on crystalline materials. For example, our previous work has proved the formation of SFLPs on a crystalline oxide material, the In₂O_{3-x}(OH)_y.^{1, 2, 3} However, for narrow band gap materials, the light penetration depth in the catalyst bed is much lower compared to our c-TiO₂@a-TiO_{2-x}(OH)_y, as amplified in Reply to Q3. Thus, the engineering of SFLP on amorphous shell is superb compared to that on narrow bandgap material in terms of light utilization.

If the "Ti_{2-x}O₂" refers to "Ti(II)O", it is prepared by reduction of TiO₂ by Ti(0) at 1500 °C, which, unfortunately, cannot be attained in our lab. If the "Ti_{2-x}O₂" refers to "TiO_{2-x}", we find that the as-prepared TiO_{2-x} also demonstrates partially amorphous surface (Figure R6), which is in agreement with previous work on defective TiO_{2-x}.^{4, 5, 6, 7, 8} Moreover, the TiO_{2-x} is much less active compared to the c-TiO₂@a-TiO_{2-x}(OH)_y towards CO₂-to-CO catalysis.

Figure R6. a. HR-TEM micrograph of H₂ reduced c-TiO₂ sample at 900 °C, 1 atm (termed c-TiO_{2-x}). b. CO production rates of c-TiO₂, H₂ reduced c-TiO₂ sample (c-TiO_{2-x}) and c-TiO₂@c-TiO_{2-x}(OH)_y, under full-spectrum Xe light (4.0 W cm⁻²).

Reference:

1. Yan T, *et al.* Polymorph selection towards photocatalytic gaseous CO₂ hydrogenation. *Nat. Commun.* 2019, **10**(1): 2521.
2. Wang L, *et al.* Room-temperature activation of H₂ by a surface frustrated lewis pair. *Angew. Chem. Int. Ed.* 2019, **58**(28): 9501-9505.
3. Ghuman KK, *et al.* Photoexcited surface frustrated lewis pairs for heterogeneous photocatalytic CO₂ reduction. *J. Am. Chem. Soc.* 2016, **138**(4): 1206-1214.
4. Naldoni A, *et al.* Effect of Nature and Location of Defects on Bandgap Narrowing in Black TiO₂ Nanoparticles. *J. Am. Chem. Soc.* 2012, **134**(18): 7600-7603.
5. Chen X, Liu L, Yu PY, Mao SS. Increasing solar absorption for photocatalysis with black hydrogenated titanium dioxide nanocrystals. *Science* 2011, **331**(6018): 746-750.
6. Naldoni A, *et al.* Photocatalysis with Reduced TiO₂: From Black TiO₂ to Cocatalyst-Free Hydrogen Production. *ACS Catal.* 2019, **9**(1): 345-364.
7. Liu X, *et al.* Progress in Black Titania: A New Material for Advanced Photocatalysis. *Adv. Energy Mater.* 2016, **6**(17): 1600452.
8. Chen X, Liu L, Huang F. Black titanium dioxide (TiO₂) nanomaterials. *Chem. Soc. Rev.* 2015, **44**(7): 1861-1885.

10. The influence of the content of the amorphous structure or the [Ti(III)-[O]- Ti(IV)-OH] bonds should be explained. If there is optimal content, the paper should present the results.

Reply: Appreciate the reviewer's comments. Based on the formation of Ti(III)-[O]-Ti(IV)-OH via the $\text{TiO}_{2-x} + 0.5y\text{H}_2\text{O} \rightarrow \text{TiO}_{2-x}(\text{OH})_y$ reaction, the amount of SFLPs corresponds to the measured x . Similar to the Reply 5, we have discussed the optical stoichiometry was about $0.0015 < x < 0.0031$ for photocatalytic CO_2 hydrogenation. Further increasing the non-stoichiometry with the thicker amorphous shell (Figure R7) will lead to electron trapping in the bulk and lowered activity in CO_2 hydrogenation (Figure R3b).

Although increasing the amount of SFLPs may create more active sites, more thicker shells will limit the charge transportation. To make full of their synergy between surface chemistry and photocatalysis, the sample $\text{c-TiO}_2 @ \text{a-TiO}_{2-x}(\text{OH})_y$ ($0.0015 < x < 0.0031$) exhibited the best performance in this work. We also added the corresponding description on page 15, line 391-406 in manuscript and the figure in Supplementary Fig. 22.

Figure R7. HR-TEM micrograph of $\text{c-TiO}_2 @ \text{a-TiO}_{2-x}(\text{OH})_y$ with increasing stoichiometry. The amorphous/crystalline interfaces are marked with dotted lines.

REVIEWERS' COMMENTS

Reviewer #1 (Remarks to the Author):

The revised manuscript has been well addressed based on the proposed comments. It could be accepted in its present form.

Reviewer #2 (Remarks to the Author):

This work investigated the interrelationship between structural disorder and surface frustrated Lewis pairs (SFLPs) in a core-shell $c\text{-TiO}_2@a\text{-TiO}_2\text{-}x(\text{OH})_y$ heterostructure for CO_2 hydrogenation photocatalysis. The results of this study are helpful for constructing efficient photocatalytic reduction of CO_2 by titanium oxide, which can provide researchers with a more in-depth understanding of the mechanism of photocatalysis. As the authors claimed in the paper, the amorphous layer plays a very significant role in this photocatalysis process. The authors demonstrated that the surface of the titanium oxide contains amorphous layers by HRTEM technology. However, the HRTEM is just a local characterization method. At the same time, the surface of titanium oxide is easily damaged by electrons and forms amorphous layers. Based on these considerations, I suggest that the authors need use other powerful methods to jointly show that the surface of titanium oxide contains amorphous layers.

REVIEWERS' COMMENTS

Reviewer #1 (Remarks to the Author):

The revised manuscript has been well addressed based on the proposed comments. It could be accepted in its present form.

Reply: We gratefully appreciate the reviewer's valuable comments and the recognition of our efforts.

REVIEWERS' COMMENTS

Reviewer #2 (Remarks to the Author):

This work investigated the interrelationship between structural disorder and surface frustrated Lewis pairs (SFLPs) in a core-shell c-TiO₂@a-TiO_{2-x}(OH)_y heterostructure for CO₂ hydrogenation photocatalysis. The results of this study are helpful for constructing efficient photocatalytic reduction of CO₂ by titanium oxide, which can provide researchers with a more in-depth understanding of the mechanism of photocatalysis. As the authors claimed in the paper, the amorphous layer plays a very significant role in this photocatalysis process. The authors demonstrated that the surface of the titanium oxide contains amorphous layers by HRTEM technology. However, the HRTEM is just a local characterization method. At the same time, the surface of titanium oxide is easily damaged by electrons and forms amorphous layers. Based on these considerations, I suggest that the authors need use other powerful methods to jointly show that the surface of titanium oxide contains amorphous layers.

Reply:

Thanks for the referee's comment. As suggested, we have utilized Raman, a powerful and surface-sensitive technique, to jointly validate the amorphous structure of the c-TiO₂@a-TiO_{2-x}(OH)_y. A typical TiO₂ Raman spectrum consists of six ($3E_g + 2B_{1g} + A_{1g}$) peaks, and pioneer work on black titania has evidenced amorphous surface on TiO₂ will cause the blue shift and broadening of the first E_g Raman peak at 145 cm⁻¹.

Based on this point, we have conducted laser power-dependent Raman measurements to both TiO₂ and c-TiO₂@a-TiO_{2-x}(OH)_y. By varying the laser power from 0.2 mW, 2 mW, 5 mW to 10 mW, the signal intensity

kept increasing while the position (145 cm^{-1}) and FWHM (13.8 cm^{-1}) of the first TiO_2 E_g peak kept constant (Figure R1a and Table R1). This ruled out the possibility of laser irradiation induced amorphous surface. As a comparison, the Raman spectra of $c\text{-TiO}_2@a\text{-TiO}_{2-x}(\text{OH})_y$ displayed a decreased signal-to-noise ratio and increased FWHM of Raman peaks compared to those of TiO_2 , and the first E_g Raman band blue-shifted to 147.5 cm^{-1} from 0.2 mW to 10 mW (Figure R1b). This result agrees with pioneer work of black titania, indicating the $c\text{-TiO}_2@a\text{-TiO}_{2-x}(\text{OH})_y$ is covered by an amorphous surface layer. To validate the reproducibility of this observation, we also carried out a $3*3$ Raman mapping using a $10x$ objective lens. The scanned area was as large as $100*100\text{ }\mu\text{m}^2$ and all the TiO_2 spectra showed E_g peak at 145 cm^{-1} (FWHM: $\sim 14\text{ cm}^{-1}$) and the $c\text{-TiO}_2@a\text{-TiO}_{2-x}(\text{OH})_y$ spectra showed the E_g peak at 147.5 cm^{-1} (FWHM: $\sim 22\text{ cm}^{-1}$), respectively. The Raman results were added in Supplementary Fig. 5.

Besides, we recorded a video during the HRTEM measurement for the $c\text{-TiO}_2@a\text{-TiO}_{2-x}(\text{OH})_y$. As shown in Figure R3, the HRTEM image showed a stable amorphous shell and a stable crystalline core during the measurement. The video was added in the Supplementary Movie 1 and the corresponding description was supplemented in the main article (line 136-140). Therefore, both Raman and HRTEM results confirmed that the amorphous layer should originate from the Na reduction instead of the structure change induced by the electron beam.

Table R1. The FWHMs of $c\text{-TiO}_2$ and $c\text{-TiO}_2@a\text{-TiO}_{2-x}(\text{OH})_y$ at the same condition.

	$c\text{-TiO}_2$	$c\text{-TiO}_2@a\text{-TiO}_{2-x}(\text{OH})_y$
10mW	13.81	22.33
5mW	13.78	21.92
2mW	13.80	21.5

Figure R1. Raman spectra of **a.** $c\text{-TiO}_2$ under different intensity of laser power. **b.** $c\text{-TiO}_2@a\text{-TiO}_{2-x}(\text{OH})_y$ under different intensity of laser power.

*10 represents multiplying the intensity by 10.

Figure R2. Raman mapping analysis of **a.** $c\text{-TiO}_2@a\text{-TiO}_{2-x}(\text{OH})_y$ and **b.** $c\text{-TiO}_2$. Right figures are the extracted Raman spectra of marked 9 sites.

Figure R3. The video of HRTEM and FFT images for $c\text{-TiO}_2@a\text{-TiO}_{2-x}(\text{OH})_y$. The video can be checked by double-click.